# Longitudinal path analysis for the directional association of depression, anxiety and posttraumatic stress disorder with their comorbidities and associated factors among postpartum women in Northwest Ethiopia: A cross-lagged autoregressive modelling study

**Marelign Tilahun Malaju[ID]¹\*, Getu Degu Alene², Telake Azale Bisetegn³**

1 Department of Public Health, College of Health Sciences, Debre Tabor University, Debre Tabor, Ethiopia,
2 School of Public Health, College of Medicine and Health Sciences, Bahir Dar University, Bahir Dar,
Ethiopia, 3 Department of Health Education and Behavioral Sciences, College of Medicine and Health
Sciences, University of Gondar, Gondar, Ethiopia

\* marikum74@gmail.com

## Abstract

### Introduction

Vulnerability for depression, anxiety and posttraumatic stress disorder symptoms due to perceived traumatic birth increase during the postpartum period. Traumatic birth has been defined as an event occurring during labour and birth that may be a serious threat to the life and safety of the mother and/or child. However, the comorbidity and multimorbidity of depression, anxiety and PTSD with their direct and indirect predictors is not well investigated in the postpartum period. In addition, the longitudinal directional association of depression, anxiety and PTSD with their comorbidities is not studied in Ethiopia.

### Objective

The aim of this study was to assess prevalence of postnatal comorbid and multimorbid anxiety, depression and PTSD. It also aimed to determine the directional association of postnatal anxiety, depression and PTSD with the comorbidity and multimorbidity of these mental health problems over time and to explore the factors that are directly or indirectly associated with comorbidity and multimorbidity of anxiety, depression and PTSD.

### Methods

A total of 775 women were included at the first, second and third follow-up of the study (6ᵗʰ, 12ᵗʰ and 18ᵗʰ week of postpartum period) during October, 2020 –March, 2021. A cross-lagged autoregressive path analysis was carried out using Stata 16.0 software in order to determine the autoregressive and cross-lagged effects of depression, anxiety and PTSD with their comorbidities. In addition, a linear structural equation modelling was also carried

Longitudinal path analysis for the directional
association of depression, anxiety and
posttraumatic stress disorder with their
comorbidities and associated factors among
postpartum women in Northwest Ethiopia: A cross-
lagged autoregressive modelling study. PLoS ONE
17(8): e0273176. https://doi.org/10.1371/journal.
pone.0273176

College Mangalore, Manipal Academy of Higher
Education, INDIA

**Data Availability Statement:** All relevant data are within the article and its Supporting Information files.

**Funding:** The author(s) received no specific funding for this work.

**Competing interests:** The authors have declared that no competing interests exist.

out to determine the direct and indirect effects of independent variables on the comorbidities of depression, anxiety and PTSD.

## Results

Comorbidity of anxiety with depression was the most common (14.5%, 12.1% and 8.1%) at the 6th, 12th and 18th week of postnatal period respectively. With regard to the direction of association, comorbidity of PTSD (due to perceived traumatic birth) with depression, PTSD with anxiety, depression with anxiety and triple comorbidity predicted depression and anxiety in subsequent waves of measurement. Direct and indirect maternal morbidity, fear of childbirth and perceived traumatic childbirth were found to have a direct and indirect positive association with comorbidities of depression, anxiety and PTSD. In contrast, higher parity, higher family size and higher social support had a direct and indirect negative association with these mental health disorders.

## Conclusion

Postnatal mental health screening, early diagnosis and treatment of maternal morbidities, developing encouraging strategies for social support and providing adequate information about birth procedures and response to mothers' needs during childbirth are essential to avert comorbidity of anxiety, depression and PTSD in the postpartum period.

## Introduction

Vulnerability for depression, anxiety and posttraumatic stress disorder symptoms increases during the postpartum period as it is a time where mothers undergo significant changes [1,2]. While depression is a state of low mood or loss of pleasure or interest in activities, anxiety is generally characterized by feelings of tension, worried thoughts and physical changes such as increased heart rate, rapid breathing, and sweating [3]. Whereas, post-traumatic stress disorder (PTSD) refers to a cluster of psychological symptoms that develop following exposure to a severe stressor or traumatic event associated with a real or perceived threat of death or threat to physical integrity of the person or others. Symptom clusters involve re-experiencing the event (intrusion symptoms), persistent avoidance of stimuli associated with the event (avoidance symptoms), negative mood alterations, and increased arousal and reactivity [3].

There have been several evidences, which have assessed prevalence of postnatal anxiety and depression in low and middle-income countries. Taken together, the findings suggest that, the prevalence rates of postnatal depression and anxiety ranged between 10 to 20% [4,5] and 8 to 17.1% respectively [5–8]. Postnatal depression was shown to be more prevalent in African research. In South Africa and Uganda, for example, 34.7 percent and 43 percent of postnatal women had depression symptoms, respectively [9,10]. It has been consistently reported by a previously published evidences that low socioeconomic status, adverse events in life, poor social support, intimate partner violence, previous history of common mental disorders, history of chronic medical illness, unmarried status and unplanned pregnancy increases the odds of perinatal anxiety and depression [11].

The prevalence of postnatal depression in Ethiopia ranged from 12.23–33.82% [12]. Domestic violence, absence of social support, marital dissatisfaction, history of infant death, unplanned pregnancy, and previous history of depression were independent determinants of

postpartum depression in Ethiopian women [12,13]. The results of a meta-analysis study in China suggests that cesarean section mode of delivery increased the risk of postpartum depression [14]. Even though the Ethiopian national cesarean section rate ranged from 2% to 29.55% [15–17], mode of delivery was not found to be associated with postpartum depression in Ethiopian study [12]. The risk factors for postpartum anxiety that have been best established are poor physical health, unplanned pregnancy, low partner support, lack of control during labor, pain, and postpartum depression [18]. Studies on perinatal affective disorders in Ethiopia have predominantly focused on postpartum depression [12] and there is no research which have investigated postnatal anxiety.

There is also a growing body of literature suggesting that women may perceive childbirth as traumatic and develop PTSD as a result of a traumatic birth [19–21]. Traumatic birth has been defined as an event occurring during labour and birth that may be a serious threat to the life and safety of the mother and/or child [22]. Several studies including systematic reviews and meta-analysis have been also conducted to determine the prevalence of postnatal PTSD symptoms. The results of these studies indicated that 9 to 45.5% of mothers perceived childbirth as traumatic [19,23,24] and PTSD prevalence rates varied from 1.8–20% [25–30].

PTSD is not the result of a single cause (i.e., a traumatic stressor), but is the consequence of various interacting variables [21]. It is the outcome of the interplay between antepartum vulnerability factors, the events during delivery and postpartum factors that interact over time during perinatal period [26]. PTSD in pregnancy can be caused by traumatic events such as accidents, interpersonal aggression, or natural disasters [27]. During the postpartum period, women can develop PTSD symptoms due to a difficult or traumatic births such as; emergency cesarean section or instrumental deliveries during which women think that they or their baby might die or be seriously hurt [31–37]. Pregnancy or childbirth complications such as hyperemesis during pregnancy and preeclampsia are associated with PTSD symptoms in the postpartum period [25,31,37–43]. Subjective birth experience (including lack of control during delivery), operative birth, low social support and dissociation were also reported as predictors of PTSD symptoms in the postpartum period [1,26,31,38]. Postpartum women may also attribute PTSD symptoms to earlier traumatic events such as childhood sexual abuse, rape or domestic violence that are not related to the perinatal period [44].

An exceptionally traumatic period due to COVID-19 pandemic marked by separation, loss of freedom [45], concern about the impact of Covid-19 on pregnancy or possible vertical transmission [46], could be a risk factor for the development of PTSD in postpartum women. There was a recommendation to separate the infant and the mother right after delivery in case of the mother's positive COVID-19 test [46]. These restrictions may lead to a lack of support during and after childbirth, as well as maternal perinatal mental health issues [46]. However, most literatures on perinatal PTSD using diagnostic measures have been conducted in high- and middle-income countries [19,28,47–50] and there is no research in this regard in Ethiopia.

The comorbidity of depression and anxiety symptoms was found to be 13.1% during the first 8 weeks postpartum [51]. Another study also reported comorbidity rates of 18.8% and 8.9% at 4–6 weeks and 6 months postpartum respectively [48]. With regard to predictors of comorbid depression and anxiety, one study reported that women with higher body mass index before pregnancy and sleep difficulties were significantly more likely to have high joint trajectories of depression and anxiety [52]. In Ethiopia, neither the status of comorbid anxiety and depression nor anxiety is reported in postpartum women but one study among pregnant women reported that the prevalence of comorbid anxiety and depression is 10.04% [11]. In this study, being married, ranked in the highest wealth quintile, having medical illness, encountering pregnancy danger signs, experiencing life-threatening events and household food insecurity were the factors which were found to be significantly associated with comorbid

anxiety and depression [11]. While the prevalence of gestational diabetes mellitus was reported to be 12.8% in Ethiopia [53], magnitude of other medical illnesses among child delivering Ethiopian women were also reported to be 40% (anemia), 5% (HIV), 3% (Tuberculosis) and 2% (Malaria) [54].

Comorbidity of PTSD with other psychiatric disorders has been also reported. While its comorbidity rate with depression was found to be 10.1% and 4.9% at 4–6 weeks and 6 months of postpartum period, it is reported that there is a comorbidity rate of 7.2% and 5.4% with anxiety during the same period [48]. In one study, women with severe childhood and postpartum trauma, negative posttraumatic cognitions, and greater dissociation were found to have a significantly greater probability of being in the chronic comorbid trajectories of depression and PTSD [1]. The rates of multimorbidity after childbirth were rarely investigated and found to be 6.6% at 4–6 weeks and 3.4% at 6 months postpartum in one study [48]. In a systematic review, the prevalence rates of multimorbidity after childbirth is reported to be 2 to 3% [55]. There is scarcity of research on the comorbidity and multimorbidity of depression, anxiety and PTSD and its predictors in the postpartum period. Only two studies were identified examining predictors of the comorbid trajectories of depression with PTSD [1] and predictors of the comorbid trajectories of depression with anxiety [52]. In addition, most previous studies were cross-sectional, providing limited insights into longitudinal directional association of depression, anxiety and PTSD with their comorbidities. Investigation of comorbidity and multimorbidity with their predictors is important, as women may present with complex and mixed symptoms, making them hard to identify and treat. To the best of the authors' knowledge, except for the prevalence of postpartum depression, there is no study which have investigated the prevalence of postpartum anxiety, PTSD, comorbidity and multimorbidity in Ethiopia.

In the light of the gaps identified, the aims of this study were: 11) to assess prevalence of postnatal comorbid and multimorbid anxiety, depression and PTSD; 2) to determine the directional association of postnatal anxiety, depression and PTSD with the comorbidity and multimorbidity of these mental health problems over time; and 3) to explore the sociodemographic, maternal, obstetric and psychosocial factors that are directly or indirectly associated with comorbidity and multimorbidity of anxiety, depression and PTSD.

## Methods and materials

### Study design and study area

As part of the health facility linked community based prospective follow-up study conducted in Northwest Ethiopia to determine the effect of maternal morbidities on maternal health related quality of life, functional status and mental health problems [56,57], postpartum women were recruited in four hospitals of south Gondar zone, Northwest Ethiopia. The data collection took place between October 1, 2020 and March 30, 2021. South Gondar is located at 650 km Northwest from Addis Ababa the capital city of Ethiopia.

### Study population

A total of 775 women consented to participate in the study and participated at the first, second and third follow-up of the study (6th, 12th and 18th week of postpartum period). Recruitment of the study participants was done after child birth and before the time of discharge within the hospitals where women gave birth. All women who were asked for consent agreed to participate in the study. The selected women were among those who were attending their antenatal care (ANC) in the study hospitals and among those who came for delivery in the study hospitals from different health centers and/or hospitals through referral. Therefore, the total

number of delivering women in the study hospitals were not known before the initiation of the study. Women differ in their socio-demographic characteristics, reproductive, obstetrics and medical variables. The effect of these variations on the outcome variables were controlled with the use of multivariable analysis in this study (multivariable linear structural equation modelling).

## Sample size determination

Sample size was calculated using a two-population proportion formula with Epi-Info version 7. Accordingly, a minimum sample size of 746 (249 exposed and 497 non-exposed) was calculated by taking 0.05 alpha (α), power of 90%, odds ratio of 4.23, 2.3% of mothers without depression during pregnancy who develop PTSD in postpartum period, 1:3 ratio of exposed to non-exposed (since the controls were 3 times the cases in a previous study [58]) and by adding 10% non-response rate. These parameters used for the sample size calculation were taken from a previous study [58].

## Eligibility/Inclusion criteria

Women aged 15 years and above, with preterm birth (between 28–36 weeks), term or post term delivery and with live birth, still birth or fetal death were included in the study. Women who are unable to communicate (having hearing problem and cannot communicate with sign language) were excluded from the study.

## Direct and indirect maternal morbidities

The direct and indirect maternal morbidities were identified based on the WHO maternal morbidity working group criteria [59]. According to the WHO maternal morbidity working group criteria, the direct maternal morbidities included in this study were: gestational hypertension, pre-eclampsia, eclampsia, placenta previa, placental abruption, postpartum hemorrhage, mastitis, puerperal sepsis, urinary tract infection, perineal tear, episiotomy wound infection, vaginal wall/perineal laceration and caesarean section wound infection. The indirect maternal morbidities included in this study based on the WHO maternal morbidity working group criteria were: asthma, tuberculosis, influenza, pneumonia, malaria, HIV/AIDS, candidiasis, hepatitis, hypertension, anemia and diabetes mellitus. Women who were diagnosed with any direct and/or indirect maternal morbidities were treated accordingly within the hospitals where they were diagnosed.

## Sampling procedure

Women diagnosed with any of the direct and indirect maternal morbidities were taken as exposed mothers and included in the study. Women without the direct and indirect maternal morbidities were taken as non-exposed mothers. All women with direct maternal morbidities were included in the study and women without direct maternal morbidities were selected by simple random sampling method using their chart number on daily bases. The chart numbers of women without direct maternal morbidities were entered into computer to generate random numbers using Microsoft Excel for random selection of women. The recruitment of women continued prospectively until the required sample size was fulfilled. Women were asked to give written consent to participate in the study and after getting their consent and full address, appointments were made at their home to collect the data for the follow up study.

## Dependent and independent variables

Comorbidity of depression, anxiety and posttraumatic stress disorder was taken as the outcome variable. Direct maternal morbidities(obstetric hemorrhage, hypertensive disorders, obstructed labour, puerperal sepsis, gestational diabetes mellitus, perineal tear), indirect maternal morbidities (anemia, malaria, hypertension, asthma, tuberculosis, HIV), socio-demographic variables (age, educational status, marital status, religion, ethnicity, occupation, monthly expenditure), residence, obstetric variables (parity, mode of delivery, gestational age at birth, birth weight, birth interval, fetal death, unwanted pregnancies, antenatal care visit, history of abortion) and psychosocial factors (social support and fear of child birth) were taken as the main independent variables.

## Measures of variables

**Depression, anxiety and stress.** The short version of depression, anxiety and stress scale 21 (DASS-21) questionnaire was used to measure depression, anxiety and stress. DASS-21 is a psychological screening instrument which is capable of differentiating symptoms of depression, anxiety and stress. It is a validated and reliable instrument with 21 items in three domains. Each domain comprises seven items assessing symptoms of depression, anxiety and stress. Participants were asked to indicate the presence of symptoms in each domain over the past week scoring from 0 (did not apply at all). to 3 (applied most of the time). Scores from each dimension were summed. Then, the final score was multiplied by 2 and then categorized according to the DASS manual as normal, mild, moderate, severe and extremely severe. Accordingly, for participants with depression, a depression score of 0–9 is considered normal, 10–13 as mild, 14–20 as moderate, 21–27 as severe and 28 and above as extremely severe. In this study a score $\geq 10$ was considered for a mother to have a symptom of depression. For participants with anxiety, an anxiety score of 0–7 is considered normal, 8–9 as mild, 10–14 as moderate, 15–19 as severe and 20 and above as extremely severe. A cut-off score of $\geq 8$ was considered to have symptoms of anxiety for this study. For participants with stress, a stress score of 0–14 is considered normal, 15–18 as mild, 19–25 as moderate, 26–33 as severe and 34 and above as extremely severe. A score of $\geq 15$ was considered to have symptoms of stress for this study. This instrument was validated and used previously in Ethiopia [60,61].

**Posttraumatic stress disorder.** The childbirth stressor was operationalized by using the Traumatic Event Scale (TES) [62,63]. In this scale, the items concerning criterion A (stressor) were formulated as follows:

1. Did you feel that the childbirth was a trying experience?

2. Did you feel that your life or your baby's life was threatened during or after the birth?

3. Did you think that you or your baby might die?

4. Did you feel anxious/helpless/horrified around the time of birth?

There were four alternative answers for each statement stated as: "not at all = 1," "somehow = 2," "much = 3," and "very much = 4." If the mother's answer is "much" or "very much" for item 1, for item 2 and item 3 or either of the two, then Criterion A is fulfilled [62,63]. The A2 criterion requiring the experience of "fear, helplessness, and horror" to qualify as a traumatic event was included in this study to examine the differences in rates of traumatic childbirth according to the DSM-IV versus DSM-V trauma criteria due to the removal of criterion A2 in DSM-V [64,65].

Following the criterion A questions, we have used the Posttraumatic Stress Disorder Checklist for DSM-5 (PCL-5) comprising the 20 PTSD symptoms (criterion B, C, D and E) to

measure PTSD over the past month. The instrument contains 20 items, including three new PTSD symptoms (compared with the PTSD Checklist for DSM-V): blame, negative emotions and reckless or self-destructive behavior [66]. A total-symptom score of zero to 80 can be obtained by summing the items. A score of 31–33 is optimal to determine PTSD symptoms and a score of ≥33 is recommended when further psychometric testing is not available [67,68]. Therefore, a score of ≥ 33 was considered to have symptoms of PTSD for this study. The cut-offs for the instrument were validated by a previous study in Ethiopia [68].

**Fear of child birth.** The Wijma Delivery Expectation/Experience Questionnaire (W-DEQ) was used to measure fear of child birth. The W-DEQ has been designed specially to measure fear of child birth operationalized by the cognitive appraisal of the delivery. This 33-item rating scale has a 6-point Likert scale as a response format, ranging from ' not at all' (= 0) to ' extremely' (= 5), yielding a score-range between 0 and 165. The Internal consistency and split-half reliability of the W-DEQ was checked in previous studies in Ethiopia with the Cronbach's alpha score of 0.932 [69,70]. A score of ≥ 85 was considered to have fear of child birth for this study [69,70].

## Social support

The Oslo 3-items social support scale with scores ranging from 3 to 14 was used to measure social support. The social support scores were categorized into poor or no social support for scores less than nine. Scores between 9 and 14 were considered moderate to strong support and merged together as "yes" for social support. The Oslo 3-items social support scale was validated and previously used in Ethiopia [71–73].

## Data collection and quality control

Administering baseline questionnaire and diagnosis of direct and indirect maternal morbidities based on the WHO criteria, were done by health professionals (BSC Nurses and Midwives) working in the Gynecology and Obstetrics wards of the study Hospitals. The questionnaire included a patient interview and record review. The interview was on socioeconomic status, medical and obstetric history and clinical symptoms. The record review was intended to extract information on selected laboratory tests and results for hemoglobin, HIV, malaria (rapid diagnostic test or smear) and glucometer (random blood sugar). The follow up data on DASS-21 and PCL-5 were administered by health extension workers (community health workers) at the first, second and third home visit (6th, 12th and 18th week of postpartum period). Two different groups of data collectors were used for data collection to avoid bias. The first group were health professionals with BSC degree in Nursing and Midwifery who collected the baseline data and the direct and indirect maternal morbidities based on the WHO maternal morbidity working group criteria. The second group were health extension workers (community health workers) who collected the follow up data by house-to-house visit (home visit). Supervision was done by the principal investigator. Training was given for data collectors in scale administration. During the training process, data collectors carefully reviewed each question and conduct pretest before the study commenced. The investigator and data collectors checked the questionnaire during the pretest and amendment was done as required.

## Data processing and analysis

We used a three-wave, cross-lagged autoregressive structural equation modeling. The analysis was carried out by using Stata version 16 software. The Autoregressive Cross-lagged (ARCL) modeling strategy was used to examine the longitudinal association of depression, anxiety and PTSD with their comorbidities including all other scales used for this study. This modeling

strategy incorporates three main components. First, the stability/autoregressive effects (e.g., effects of T1 comorbid depressive, anxiety and PTSD symptoms on their respective T2 variables). That means, later measures of a construct are predicted by earlier measures of the same construct. Second, the cross-lagged effects (e.g., effect of T1 depressive symptoms on T2 comorbid PTSD and anxiety symptoms and of T1 comorbid PTSD and anxiety symptoms on T2 depressive symptoms). That means, earlier measures of depression predict later measures of comorbid PTSD and anxiety symptoms. This model can be extended to examine bi-directional relations such that earlier measures of PTSD predict later measures of comorbid depression and anxiety as well. Third, the synchronous associations between the unexplained variances of these variables at T1, T2 and T3 [74,75]. We estimated the model fitness by using the comparative fit index (CFI), Tucker-Lewis Index (TLI) and the root-mean-square error of approximation (RMSEA) based on the Satorra-Bentler correction. Both the TLI and CFI should be greater than 0.90, but the RMSEA value should be less than 0.08 to judge the model as reasonably fitting the data [21,74]. The aim of the analysis was to examine the cross-lagged and autoregressive association of depressive, anxiety and PTSD with their comorbidities, controlling for the confounder variables.

As this study involved repeated measurements, we have carried out test of temporal invariance (homogeneity) for depression, anxiety, PTSD and their comorbidities across the three data points in time [76]. Homogeneity is to mean that statistical properties of any one part of an overall dataset are the same as any other part. Therefore, homogeneity of paths was conducted to test the regression coefficients for each observed variable is stable over time. It can be tested by comparing if the impact of the variable (regression coefficient) at the time of [t-1] on the variable at the time of [t] is the same with the impact of the variable (regression coefficient) at the time of [t] on the variable at the time of [t+1]. The regression coefficient can be labeled as stability coefficient [77,78]. Accordingly, Model 1 served as the base model with no invariance constraints (freely estimated factor loadings, autoregressive paths, cross lagged paths and residual covariance at a different time point) to test configural invariance. In Model 2, we constrained the auto-regressive and cross-lagged paths for depression, anxiety, PTSD and their comorbidities across time to be equal. We then compared the model fit indices of the two models to select the best model. Test of difference in CFI, TLI and RMSEA (ΔCFI, ΔTLI, ΔRMSEA) were used to compare the fit of the nested models. If a difference of less than 0.01 is found in the ΔCFI, ΔTLI and ΔRMSEA index between nested models, it is indicated that the more constrained model should be supported [79–81].

In addition, the direct and indirect relationships between the independent and dependent variables was also explored using a linear structural equation modeling. This allowed us to assess the strength of the hypothesized direct and indirect causal pathways. Estimated effects for which $p < 0.05$ were considered as being statistically significant. The direct effect is the pathway from the exogenous (independent) variable to the outcome (endogenous) variable while controlling for the mediator. Therefore, in the indicated path diagram of Fig 1 bellow, y and z are the direct effects of anxiety and depression on PTSD respectively [82]. The indirect effect describes the pathway from the exogenous (independent) variable to the outcome (endogenous) variable through the mediator. This path is represented through the product of x and y in the path diagram of Fig 1 bellow [82]. This indicates the indirect effect of depression on PTSD through anxiety which is the mediator between depression and PTSD. Finally, the total effect is the sum of the direct and indirect effects of the exogenous (independent) variables on the outcome (endogenous) variable, Z + XY [82].

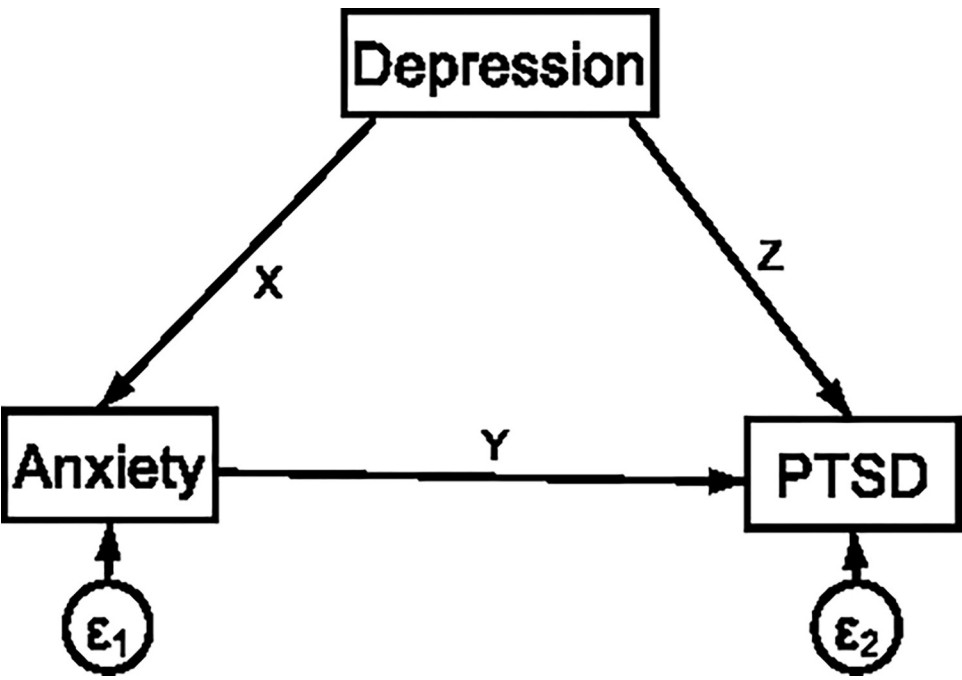

**Fig 1. Example of a mediation analysis pathway for the direct and indirect (through anxiety) association of depression with PTSD symptom.**

## Ethical considerations

Ethical approval was obtained from Institutional Review Board of Bahir Dar University (Reference number: 00225/2020). All institutions of the study area were also approached for permission. Each study participant has given written informed consent to participate in the study before taking part after giving birth and before discharge. Written informed consent for women of less than 18 years was obtained from their guardians or care givers. In addition to written informed consent from their guardians or care givers, assent was also obtained from teenage mothers whose age is less than 18 years, to participate in the study. The written informed consent and assent were taken after clear explanation about the purpose of the study. The study participants and guardians or care givers of women whose age is less than 18 years were informed that they have full right to withdraw from the study without losing any of their right as a client in the health institutions. Data collection (scale administration) at hospitals took place in a private room of delivery wards and the follow up data collection at household were also carried out in a private place within the mothers' living compound. Using codes, passwords and limiting access to the data only for the investigators were the measures taken to ensure the confidentiality of the data. Data collectors read out and assisted participants to fill out the consent form if participants were unable to read and write. This was witnessed by a neutral observer to prevent coercion.

## Results

Despite the fact that the predicted sample size was 746, a total of 779 postpartum women were included at baseline, in the hopes of improving the study's results and reducing the impacts of loss to follow up. Out of the 779 women recruited at baseline, 775(99.5%) of them participated at the first, second and third follow-up of the study (6th, 12th and 18th week of postpartum

period). Four mothers were lost to follow up because of changing their place of living and going out of the study area. The mean age of the study participants was 26.3(4.36). Almost all of them 774(99.9%) were Amhara by ethnicity and 742(95.7%) were followers of Orthodox Christianity. Other socio-demographic characteristics of mothers are shown in Table 1.

## Individual prevalence rates of depression, anxiety and PTSD symptoms

The prevalence of depression, anxiety and PTSD symptoms at the 6th, 12th and 18th week of postpartum period was computed and is provided in Table 2. The most common disorder was anxiety followed by depression throughout the follow up period. PTSD symptom was rarely reported at each time point. Stressor criteria A for a traumatic birth were fulfilled by 37.03% of women using DSM-IV criteria and 39.5% using DSM-5 criteria. Using DSM-5 stressor criteria therefore increased the number of women identified as fulfilling stressor criteria by 2.47%.

## Prevalence of comorbid depression, anxiety and PTSD in the postpartum period

The prevalence of comorbid depression, anxiety and PTSD at the 6th, 12th and 18th week of postpartum period was computed and presented in Table 3. Comorbidity of anxiety and depression was the most common. PTSD demonstrated almost the same comorbidity with

**Table 1. Socio-demographic characteristics of postpartum women in Northwest Ethiopia, 2021.**

| Variables | Total n (%) |
|---|---|
| **Age** [Mean(±SD) = 26.33(±4.355)] | |
| **Residence** Urban Rural | 771(99.5) 4(0.5) |
| **Ethnicity** Amhara Tigre | 774 (99.9) 1 (0.1) |
| **Religion** Orthodox Muslim Protestant | 742 (95.7) 30 (3.9) 3 (0.4) |
| **Education status** Illiterate/read and write Grade 1–8 Grade 9–12 Certificate/Diploma Degree and higher | 65 (8.4) 136 (17.5) 219 (28.3) 217 (28.0) 138 (17.8) |
| **Occupation** Gov't employed Merchant/Student Housewife Farmer/Daily laborer | 230 (29.7) 145 (18.7) 367 (47.4) 33 (4.3) |
| **Marital Status** Married Single/widowed/divorced | 762 (98.3) 13 (1.7) |
| **Monthly expenditure** < = 3000 Ethiopian currency 3001–4000 Ethiopian currency > = 4001 Ethiopian currency | 206 (26.6) 192 (24.8) 377 (48.6) |

**Table 2. Prevalence of depression, anxiety and PTSD among postpartum women in Northwest Ethiopia, 2021.**

| Follow up period | Type of mental health disorder | | | | | | Fulfill DSM-5 criterion A | | Fulfill DSM-IV criterion A | |
|---|---|---|---|---|---|---|---|---|---|---|
| | PTSD | | Depression | | Anxiety | | | | | |
| | n (%) | 95% CI | n (%) | 95% CI | n (%) | 95% CI | n (%) | 95% CI | n (%) | 95% CI |
| 6th week | 75(9.7%) | 7.8, 12.0 | 120(15.5%) | 13.1, 18.2 | 143(18.5%) | 15.9,21.3 | 306(39.5) | 36.1, 43.0 | 287 (37.03) | 34.3, 41.1 |
| 12th week | 53(6.8%) | 5.3, 8.9 | 100(12.9%) | 10.7, 15.5 | 120(15.5%) | 13.1,18.2 | | | | |
| 18th week | 27(3.5%) | 2.4, 5.0 | 67(8.6%) | 6.9,10.8 | 74(9.5%) | 7.7, 11.8 | | | | |
| Total number of women | 775 | | 775 | | 775 | | 775 | | 775 | |

depression and with anxiety at all time points. The triple morbidity was most notable at the first follow up time.

## Stability and cross-lagged association of depression, anxiety and PTSD with comorbidity of depression and anxiety

In order to assess the stability and cross-legged association of depression, anxiety and PTSD with the co-morbidity of depression and anxiety, a cross-lagged autoregressive analysis was carried out and the result was presented in Fig 2 and Table 4. We have tested the temporal invariance of the structural model for the longitudinal association of depression, anxiety and PTSD with comorbidity of depression and anxiety across three data points in time. A constrained model where the autoregressive and cross-lagged path coefficients were constrained to be equal across three time points did not significantly differ from the unconstrained model where the parameters were freely estimated as evidenced by the fit indices test of difference ($\Delta$CFI = 0.005, $\Delta$TLI = 0.001, $\Delta$RMSEA = 0). Model fit for the unconstrained model based on Satorra-Bentler correction: CFI_SB = 0.983, TLI_SB = 0.964, SRMR = 0.045 and RMSEA_SB = 0.042. Model fit for the constrained model based on Satorra-Bentler correction: CFI_SB = 0.981, TLI_SB = 0.965, SRMR = 0.086 and RMSEA_SB = 0.042. Therefore, the results indicated that depression, anxiety, PTSD and comorbidity of depression with anxiety showed factorial invariance across the three waves and as a result we have used the constrained model for this study.

The autoregressive analyses revealed that comorbidity of depression and anxiety were stable over time which indicates that participants suffering from comorbidity of depression and anxiety in Time 1 continue to suffer from this comorbidity over time. Comorbidity of anxiety with

**Table 3. Prevalence of Comorbid depression, anxiety and PTSD among postpartum women in Northwest Ethiopia, 2021.**

| Follow up period | Type of mental health disorder | | | | | | | |
|---|---|---|---|---|---|---|---|---|
| | PTSD and depression | | PTSD and Anxiety | | Anxiety and depression | | Multimorbidity | |
| | n (%) | 95% CI | n (%) | 95% CI | n (%) | 95% CI | n (%) | 95% CI |
| 6th week | 73 (9.4%) | 7.6, 11.7 | 72 (9.3%) | 7.4, 11.6 | 112 (14.5%) | 12.1, 17.1 | 72 (9.3%) | 7.4, 11.6 |
| 12th week | 51 (6.6%) | 5.0, 8.6 | 50 (6.5%) | 4.9, 8.4 | 94 (12.1%) | 10.0, 14.6 | 50 (6.5%) | 4.9, 8.4 |
| 18th week | 23 (3.0%) | 2.0, 4.4 | 23 (3.0%) | 2.0, 4.4 | 63 (8.1%) | 6.4, 10.3 | 23 (3.0%) | 2.0, 4.4 |
| Total number of women | 775 | | 775 | | 775 | | 775 | |

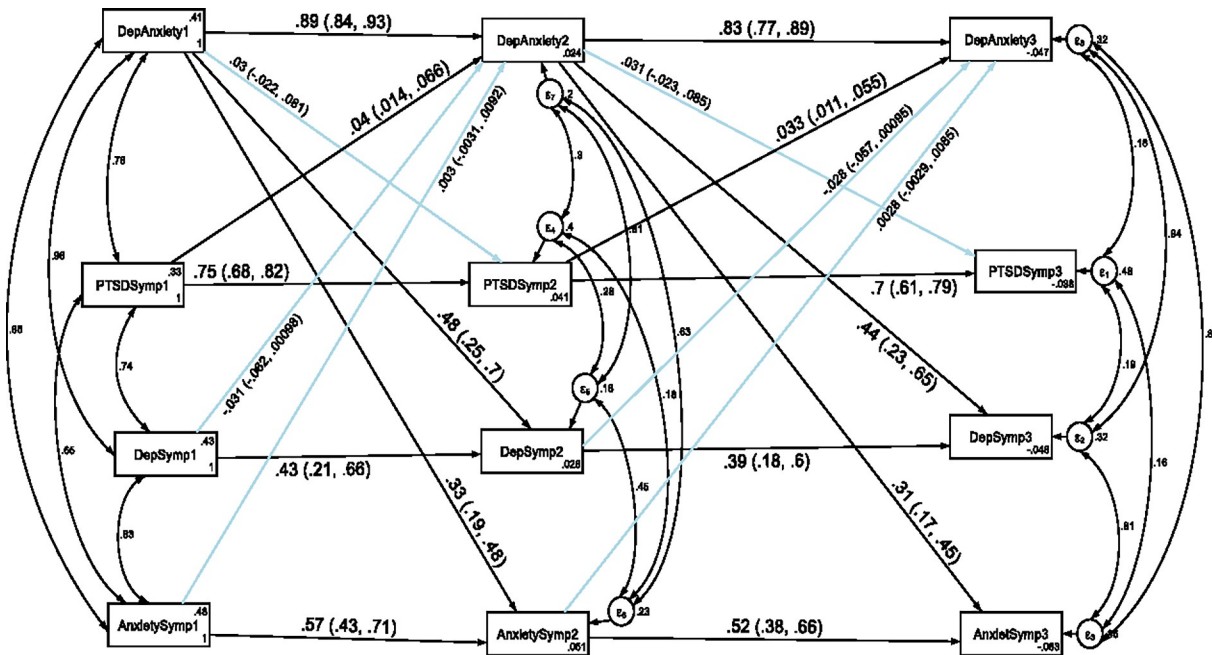

**Fig 2. Longitudinal stability and cross-lagged association of depression, anxiety and PTSD with comorbidity of depression and anxiety.**
*Note*: All β's are standardized estimates with 95% CI.

depression at T1 was associated with T2 comorbid anxiety and depression, and this pattern was extended to T3. With regard to the cross-lagged effect, co-morbidity of depression with anxiety at T1 predicted depression and anxiety symptoms at T2 and co-morbidity of depression with anxiety at T2 predicted anxiety symptoms at T3. However, suffering from depression or anxiety symptoms did not change the likelihood of co-morbidity between depression and anxiety symptoms in subsequent waves of measurement (see Fig 2 and Table 4).

## Stability and cross-lagged association of depression, anxiety and PTSD with comorbidity of depression and PTSD

The results of another cross-lagged autoregressive analysis for the stability and cross-legged association of depression, anxiety and PTSD with the co-morbidity of depression and PTSD, were presented in Fig 3 and Table 4. We have also tested the temporal invariance of the structural model for the longitudinal association of depression, anxiety and PTSD with comorbidity of depression and PTSD across three data points in time. A constrained model where the autoregressive and cross-lagged path coefficients were constrained to be equal across three time points did not significantly differ from the unconstrained model where the parameters were freely estimated as evidenced by the fit indices test of difference ($\Delta$CFI = 0.002, $\Delta$TLI = 0.012, $\Delta$RMSEA = 0.005). Model fit for the unconstrained model based on Satorra-Bentler correction: CFI_SB = 0.972, TLI_SB = 0.940, SRMR = 0.031 and RMSEA_SB = 0.044. Model fit for the constrained model based on Satorra-Bentler correction: CFI_SB = 0.970, TLI_SB = 0.952, SRMR = 0.081 and RMSEA_SB = 0.039. Therefore, the results indicated that depression, anxiety, PTSD and comorbidity of depression and PTSD showed factorial invariance across the three waves and we have used the constrained model for this study.

The autoregressive path coefficients revealed that comorbidity of depression and PTSD predicted subsequent comorbidity itself. The cross-lagged path coefficients indicated that,

**Table 4. Results of a cross-lagged autoregressive analysis for the association of depression, anxiety and PTSD with their comorbidities, among postpartum women in Northwest Ethiopia, 2021.**

| The autoregressive analysis between comorbidities of depression, anxiety and PTSD | | | |
|---|---|---|---|
| **The Autoregressive effect results** | **Standardized β (95%CI)** | **SE** | **P-value** |
| **Prediction of:** | | | |
| T2 Depression & PTSD comorbidity by T1 comorbidity | 0.75(0.57, 0.92) | 0.09 | <0.001 |
| T3 Depression & PTSD comorbidity by T2 comorbidity | 0.69(0.52, 0.85) | 0.08 | <0.001 |
| T2 Anxiety & PTSD comorbidity by T1 comorbidity | 0.76(0.62, 0.90) | 0.07 | <0.001 |
| T3 Anxiety & PTSD comorbidity by T2 comorbidity | 0.71(0.58, 0.84) | 0.07 | <0.001 |
| T2 Depression & anxiety comorbidity by T1 comorbidity | 0.89(0.83, 0.95) | 0.03 | <0.001 |
| T3 Depression & anxiety comorbidity by T2 comorbidity | 0.83(0.77, 0.89) | 0.03 | <0.001 |
| T2 Multimorbidity by T1 Multimorbidity | 0.86(0.67, 1.04) | 0.09 | <0.001 |
| T3 Multimorbidity by T2 Multimorbidity | 0.63(0.42, 0.83) | 0.10 | <0.001 |
| **The cross-lagged analysis between comorbidities of depression, anxiety and PTSD** | | | |
| **The Cross-lagged effect results** | **Standardized β (95%CI)** | **SE** | **P-value** |
| **Prediction of:** | | | |
| T2 Depression by T1 PTSD & depression comorbidity | 0.26(0.23, 0.30) | 0.02 | <0.001 |
| T3 Depression by T2 PTSD & depression comorbidity | 0.22(0.19, 0.25) | 0.02 | <0.001 |
| T2 anxiety by T1 PTSD & depression comorbidity | 0.27(0.24, 0.31) | 0.02 | <0.001 |
| T3 anxiety by T2 PTSD & depression comorbidity | 0.23(0.20, 0.26) | 0.02 | <0.001 |
| T2 Depression by T1 PTSD & anxiety comorbidity | 0.27(0.23, 0.30) | 0.02 | <0.001 |
| T3 Depression by T2 PTSD & anxiety comorbidity | 0.23(0.19, 0.26) | 0.02 | <0.001 |
| T2 anxiety by T1 PTSD & anxiety comorbidity | 0.27(0.24, 0.31) | 0.02 | <0.001 |
| T3 anxiety by T2 PTSD & anxiety comorbidity | 0.23(0.20, 0.26) | 0.02 | <0.001 |
| T2 anxiety by T1 depression & anxiety comorbidity | 0.33(0.29, 0.38) | 0.02 | <0.001 |
| T3 anxiety by T2 depression & anxiety comorbidity | 0.31(0.27, 0.35) | 0.02 | <0.001 |
| T2 depression by T1 depression & anxiety comorbidity | 0.48(0.41, 0.55) | 0.04 | <0.001 |
| T3 depression by T2 depression & anxiety comorbidity | 0.44(0.37, 0.50) | 0.03 | <0.001 |
| T2 depression by T1 triple comorbidity | 0.21(0.17, 0.25) | 0.02 | <0.001 |
| T3 depression by T2 triple comorbidity | 0.28(0.23, 0.33) | 0.03 | <0.001 |
| T2 anxiety by T1 triple comorbidity | 0.19(0.15, 0.23) | 0.02 | <0.001 |
| T3 anxiety by T2 triple comorbidity | 0.30(0.25, 0.34) | 0.03 | <0.001 |

comorbidity of depression and PTSD at T1, predicted depression and anxiety at T2. Comorbidity of depression and PTSD at T2, also predicted depression and anxiety at T3. However, depression, anxiety and PTSD did not significantly predict comorbidity of depression and PTSD at different cross-time points.

## Stability and cross-lagged association of depression, anxiety and PTSD with comorbidity of anxiety and PTSD

To examine the stability and cross-legged association of depression, anxiety and PTSD with co-morbidity of anxiety and PTSD, another cross-lagged autoregressive analysis was also carried out and the results were presented in Fig 4 and Table 4. We have also tested the temporal invariance of the structural model for the longitudinal association of depression, anxiety and PTSD with comorbidity of anxiety and PTSD across three data points in time. A constrained model where the autoregressive and cross-lagged path coefficients were constrained to be equal across three time points did not significantly differ from the unconstrained model where the parameters were freely estimated as evidenced by the fit indices test of difference

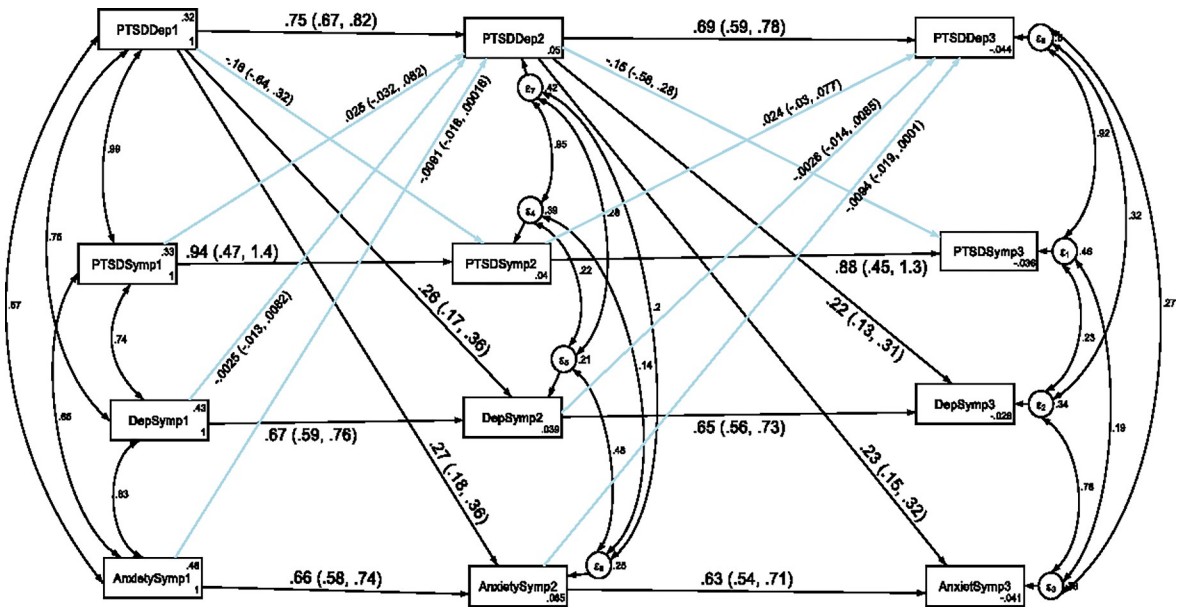

**Fig 3. Longitudinal stability and cross-lagged association of depression, anxiety and PTSD with comorbidity of depression and PTSD.** *Note*: All β's are standardized estimates with 95% CI.

(ΔCFI = 0.004, ΔTLI = 0.017, ΔRMSEA = 0.004). Model fit for the unconstrained model based on Satorra-Bentler correction: CFI_SB = 0.981, TLI_SB = 0.959, SRMR = 0.031 and RMSEA_SB = 0.035. Model fit for the constrained model based on Satorra-Bentler correction: CFI_SB = 0.985, TLI_SB = 0.976, SRMR = 0.079 and RMSEA_SB = 0.027. Hence, the results indicated that depression, anxiety, PTSD and comorbidity of anxiety and PTSD showed factorial invariance across the three waves and the constrained model was used for this study.

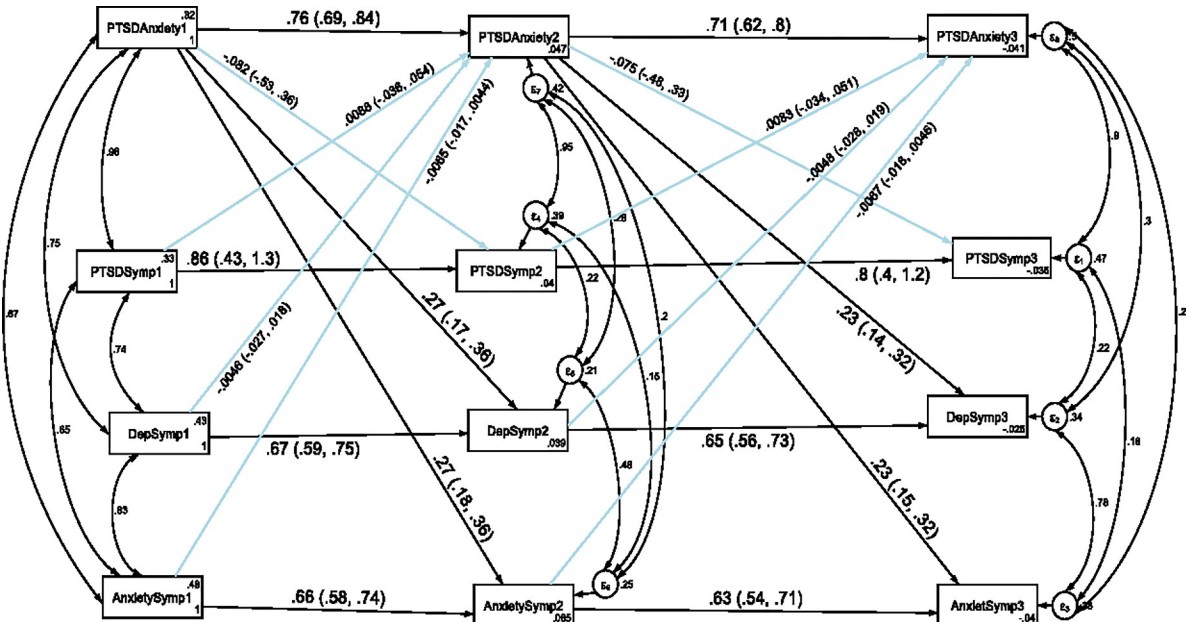

**Fig 4. Longitudinal stability and cross-lagged association of depression, anxiety and PTSD with comorbidity of anxiety and PTSD.** *Note*: All β's are standardized estimates with 95% CI.

The autoregressive path coefficients revealed that comorbidity of anxiety and PTSD predicted subsequent comorbidity itself. The cross-lagged path coefficients indicated that, comorbidity of anxiety and PTSD at T1, predicted depression and anxiety at T2. Comorbidity of anxiety and PTSD at T2, also predicted depression and anxiety at T3. However, depression, anxiety and PTSD did not significantly predict comorbidity of anxiety and PTSD at different cross-time points.

## Stability and cross-lagged association of depression, anxiety and PTSD with multimorbidity

We have also carried out a cross-lagged autoregressive analysis to examine whether anxiety, depression and PTSD predict triple comorbidity and the results were presented in Fig 5 and Table 4. We have tested the temporal invariance of the structural model for the longitudinal association of depression, anxiety and PTSD with triple comorbidity across three data points in time. A constrained model where the autoregressive and cross-lagged path coefficients were constrained to be equal across three time points did not significantly differ from the unconstrained model where the parameters were freely estimated as evidenced by the fit indices test of difference (ΔCFI = 0.004, ΔTLI = 0.017, ΔRMSEA = 0.004).

Model fit for the unconstrained model based on Satorra-Bentler correction: CFI_SB = 0.981, TLI_SB = 0.959, SRMR = 0.031 and RMSEA_SB = 0.035. Model fit for the constrained model based on Satorra-Bentler correction: CFI_SB = 0.985, TLI_SB = 0.976, SRMR = 0.079 and RMSEA_SB = 0.027. Hence, the results indicated that depression, anxiety, PTSD and triple comorbidity showed factorial invariance across the three waves and the constrained model was used for this study.

The autoregressive path coefficients indicated that triple comorbidity predicted subsequent triple comorbidity itself. This indicates that participants who suffered from triple comorbidity at T1 tended to suffer from full comorbidity both at T2 and T3. The cross-lagged path coefficients also indicated that, triple comorbidity at T1, predicted depression and anxiety at T2. Triple comorbidity at T2, also predicted depression and anxiety at T3. However, depression, anxiety and PTSD did not significantly predict triple comorbidity at different cross-time points.

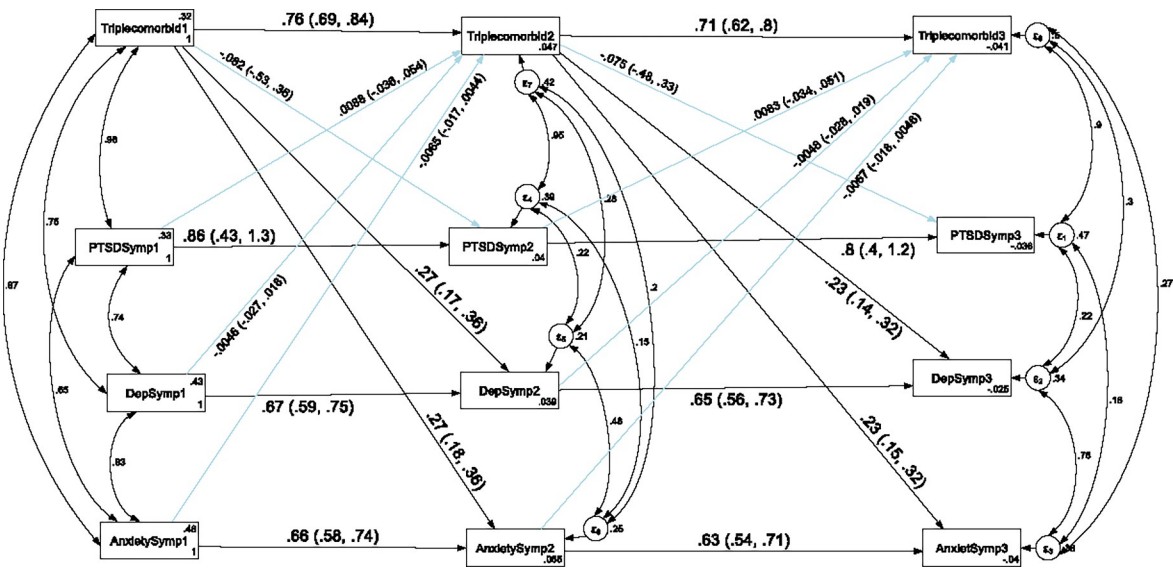

**Fig 5. Longitudinal stability and cross-lagged association of depression, anxiety and PTSD with multimorbidity of anxiety, depression and PTSD.** *Note*: All β's are standardized estimates with 95% CI.

In sum, while comorbidity of depression with anxiety predicted depression only from T1 to T2, it predicts anxiety in subsequent waves of measurement. In addition, comorbidity of PTSD with depression, PTSD with anxiety and triple comorbidity predicted depression and anxiety in subsequent waves of measurement.

### Direct, indirect and total effects of variables associated with comorbidity of anxiety and depression at each follow up period

The structural equation model for comorbidity of depression and anxiety fitted the data well according to various fit indices (CFI = 0.976, TL = 0.908, RMSEA = 0.096 and SRMR = 0.038). Direct and indirect maternal morbidity, fear of childbirth, multigravidity and perceived traumatic birth were the factors directly associated with comorbidity of depression and anxiety throughout the follow up period. In addition, fear of childbirth and multigravidity had also an indirect positive effect on comorbidity of depression and anxiety symptoms at the first, second and third follow up period through perceived traumatic birth.

In contrast, multiparity and higher social support had a direct negative effect on comorbidity of depression and anxiety symptoms at the first(T1), second(T2) and third(T3) follow up period. Moreover, higher social support had an indirect negative effect on comorbidity of depression and anxiety symptoms at the first, second and third follow up period through perceived traumatic birth (see Fig 6 and Table 5).

### Direct, indirect and total effects of variables associated with comorbidity of anxiety and PTSD at each follow up period

The structural equation model for comorbidity of anxiety and PTSD also fitted the data well according to various fit indices (CFI = 0.968, TL = 0.874, RMSEA = 0.096 and SRMR = 0.033).

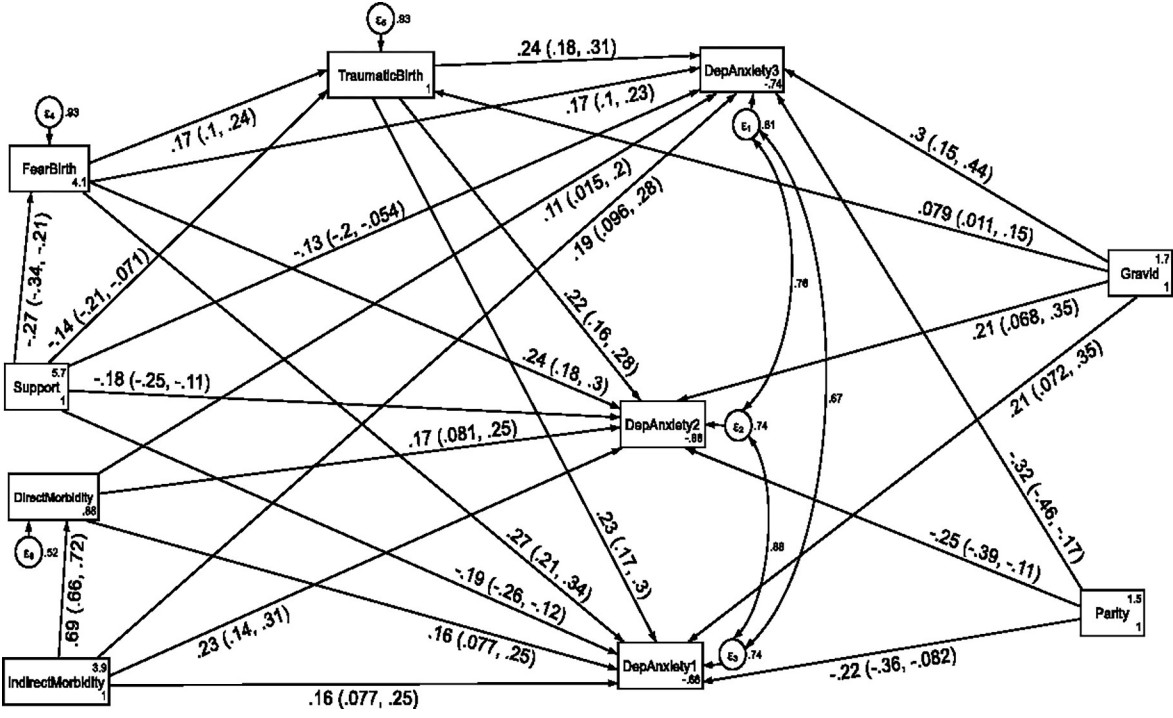

**Fig 6. A structural equation model of the factors associated with comorbidity of depression and anxiety symptoms in postpartum women, Northwest Ethiopia.** Note: β's are standardized estimates with 95% CI.

**Table 5. Direct, Indirect and total effects of variables associated with comorbidity of depression with anxiety, among postpartum women in Northwest Ethiopia, 2021.**

| Variable's pathway | Type of mental health disorder at each follow up period | | | | | | | | |
| --- | --- | --- | --- | --- | --- | --- | --- | --- | --- |
| | T1 anxiety & depression | | | T2 anxiety & depression | | | T3 anxiety & depression | | |
| | Direct effect | Indirect effect | Total effect | Direct effect | Indirect effect | Total effect | Direct effect | Indirect effect | Total effect |
| | β (SE) | β (SE) | β (SE) | β (SE) | β (SE) | β (SE) | β (SE) | β (SE) | β (SE) |
| Direct morbidity Yes (252) | 0.12(0.03) [a] | No path | 0.12(0.03) [a] | 0.12(0.03) [a] | No path | 0.12(0.03) [a] | 0.06(0.03) [b] | No path | 0.06(0.03) [b] |
| Fear of childbirth Yes (245) | 0.004(0.001) [a] | 0.001 (0.0001) [a] | 0.01(0.001) [a] | 0.003 (0.0004) [a] | 0.001(0.0001) [a] | 0.004(0.0004) [a] | 0.002(0.0004) [a] | 0.001(0.0001) [a] | 0.002(0.0004) [a] |
| Gravidity | 0.07(0.02) [b] | 0.01(0.003) [b] | 0.07(0.02) [a] | 0.06(0.02) [b] | 0.01(0.002) [b] | 0.07(0.02) [b] | 0.07(0.02) [a] | 0.01(0.002) [b] | 0.08(0.02) [a] |
| Parity | -0.07(0.02) [b] | No path | -0.07(0.02) [b] | -0.08(0.02) [a] | No path | -0.08(0.02) [a] | -0.08(0.02) [a] | No path | -0.08(0.02) [a] |
| Social support | -0.03(0.01) [a] | -0.02(0.003) [a] | -0.06(0.01) [a] | -0.03(0.01) [a] | -0.02(0.003) [a] | -0.05(0.01) [a] | -0.02(0.01) [a] | -0.01(0.002) [a] | -0.03(0.005) [a] |
| Traumatic birth Yes (306) | 0.17(0.02) [a] | No path | 0.17(0.02) [a] | 0.15(0.02) [a] | No path | 0.20(0.02) [a] | 0.14(0.02) [a] | No path | 0.14(0.019) [a] |
| Indirect morbidity Yes (210) | 0.13(0.04) [a] | 0.09(0.03) [a] | 0.22(0.03) [a] | 0.17(0.03) [a] | 0.09(0.02) [a] | 0.30(0.03) [a] | 0.12(0.03) [a] | 0.05(0.02) [b] | 0.16(0.02) [a] |

[a] p-value ≤0.001

[b] p-value <0.01, β is standard estimate.

As shown in Fig 7 and Table 6, indirect maternal morbidity, fear of childbirth, and perceived traumatic birth were the factors directly associated with comorbidity of anxiety and PTSD throughout the follow up period. Direct maternal morbidity had a direct positive effect on comorbidity of anxiety and PTSD at the first and second follow up period.

In addition, indirect maternal morbidity (asthma, tuberculosis, pneumonia, hypertension, anemia and diabetes mellitus) had also an indirect positive effect on comorbidity of anxiety and PTSD symptoms at the first and second follow up period through direct maternal morbidity. In contrast, higher social support had an indirect negative effect on comorbidity of anxiety and PTSD symptoms at the first, second and third follow up period through perceived traumatic birth (see Fig 7 and Table 6).

## Direct, indirect and total effects of variables associated with comorbidity of depression and PTSD at each follow up period

The structural equation model for comorbidity of depression and PTSD fitted the data well according to various fit indices (CFI = 0.968, TL = 0.874, RMSEA = 0.096 and SRMR = 0.033). Comorbidity of depression and PTSD symptoms was found to have a direct positive association with indirect maternal morbidity and perceived traumatic birth throughout the follow up period. Direct maternal morbidity had a direct positive effect on the comorbidity of depression and PTSD symptoms only at the first follow up period. In addition, fear of childbirth had also an indirect positive association with comorbidity of depression and PTSD symptoms at the first and second follow up period through perceived traumatic birth.

In contrast, higher social support had also a direct negative effect on comorbidity of depression and PTSD symptoms at the first(T1) and third(T3) follow up period and an indirect

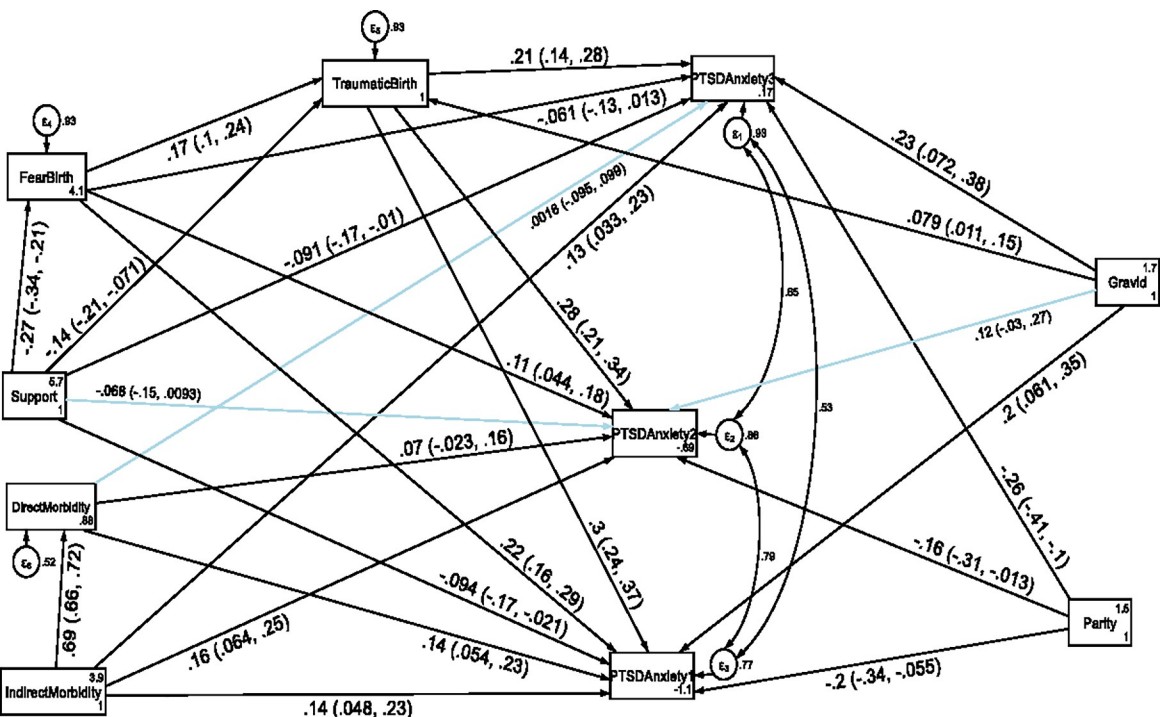

**Fig 7. A structural equation model of the factors associated with comorbidity of anxiety and PTSD symptoms in postpartum women, Northwest Ethiopia.** Note: β's are standardized estimates with 95% CI.

**Table 6. Direct, Indirect and total effects of variables associated with comorbidity of anxiety with PTSD, among postpartum women in Northwest Ethiopia, 2021.**

| Variable's pathway | Type of mental health disorder at each follow up period | | | | | | | | |
|---|---|---|---|---|---|---|---|---|---|
| | T1 Anxiety & PTSD | | | T2 Anxiety & PTSD | | | T3 Anxiety & PTSD | | |
| | Direct effect | Indirect effect | Total effect | Direct effect | Indirect effect | Total effect | Direct effect | Indirect effect | Total effect |
| | β (SE) | β (SE) | β (SE) | β (SE) | β (SE) | β (SE) | β (SE) | β (SE) | β (SE) |
| Direct morbidity Yes (252) | 0.09(0.03) [b] | No path | 0.09(0.03) [b] | 0.04(0.03) [c] | No path | 0.04(0.03) [c] | 0.001(0.02) [c] | No path | 0.001(0.02) [c] |
| Fear of childbirth Yes (245) | 0.003 (0.0004) [a] | 0.001 (0.0001) [a] | 0.003 (0.0004) [a] | 0.001 (0.0003) [b] | 0.001 (0.0001)[a] | 0.002 (0.0003) [a] | -0.0004 (0.0002)[c] | 0.0002 (0.0001) [a] | -0.0002 (0.0002) [c] |
| Gravidity | 0.05(0.02) [b] | 0.01(0.003)[b] | 0.06(0.02) [b] | 0.03(0.02)[c] | 0.01(0.002)[b] | 0.03(0.02)[c] | 0.04(0.01)[b] | 0.002(0.001)[b] | 0.04(0.01)[b] |
| Parity | -0.05(0.02) [b] | No path | -0.05(0.02) [b] | -0.04(0.02)[b] | No path | -0.04(0.02)[b] | -0.04(0.01)[a] | No path | -0.04(0.01)[a] |
| Social support | -0.02(0.01)[b] | -0.02(0.003)[a] | -0.03(0.01) [a] | -0.01(0.005)[c] | -0.01(0.002)[a] | -0.02(0.005)[a] | 0.01(0.004)[b] | 0.002(0.001)[c] | 0.01(0.003)[b] |
| Traumatic birth Yes (306) | 0.18(0.02)[a] | No path | 0.18(0.02)[a] | 0.14(0.02)[a] | No path | 0.14(0.02)[a] | 0.07(0.01)[a] | No path | 0.07(0.01)[a] |
| Indirect morbidity Yes (210) | 0.09(0.03) [b] | 0.07(0.02)[b] | 0.16(0.02) [a] | 0.09(0.03)[a] | 0.03(0.018) [c] | 0.11(0.02) [a] | 0.05(0.02)[b] | 0.001(0.013)[c] | 0.05(0.02)[a] |

[a] p-value ≤0.001

[b] p-value <0.01

[c] p-value <0.05, β is standard estimate.

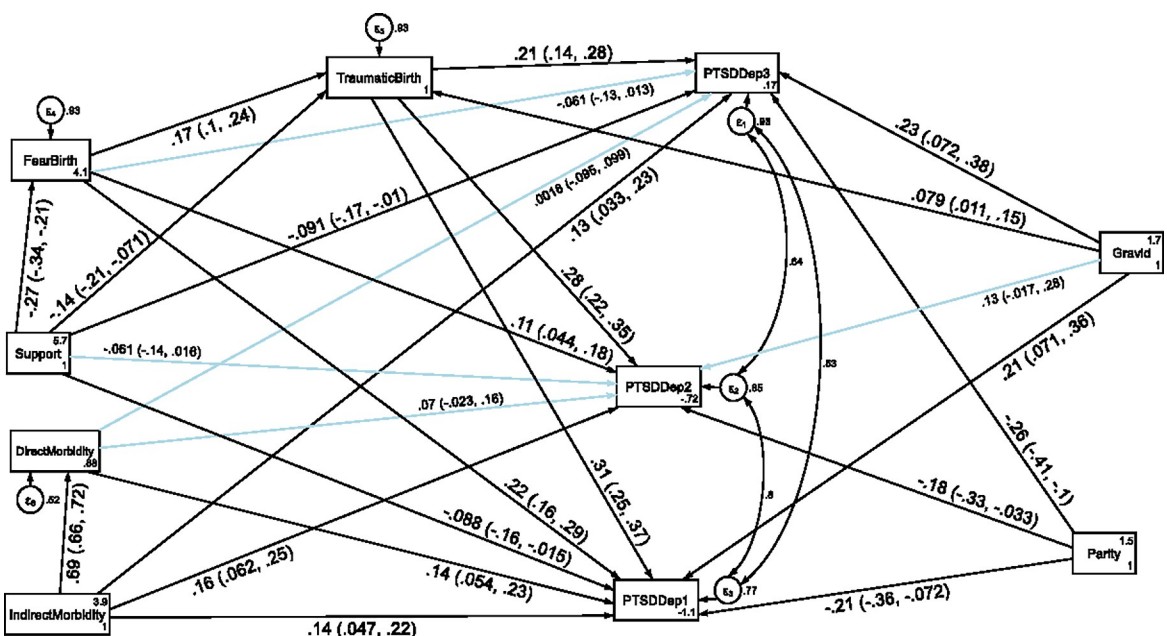

**Fig 8. A structural equation model of the factors associated with comorbidity of depression and PTSD symptoms in postpartum women, Northwest Ethiopia.** Note: β's are standardized estimates with 95% CI.

negative effect through perceived traumatic birth at the same follow up period (see Fig 8 and Table 7).

## Direct, indirect and total effects of variables associated with multimorbidity at each follow up period

The structural equation model for triple comorbidity fitted the data well according to various fit indices (CFI = 0.968, TL = 0.874, RMSEA = 0.096 and SRMR = 0.033). Triple comorbidity was found to have a direct positive association with indirect maternal morbidity and perceived traumatic birth throughout the follow up period. In addition, fear of childbirth had an indirect positive association with triple comorbidity symptoms at the first and second follow up period through perceived traumatic birth.

On the contrary, higher social support had a direct negative effect on triple comorbidity symptoms at the first(T1) and third(T3) follow up period and an indirect negative effect through perceived traumatic birth at the same follow up period (see Fig 9 and Table 8).

## Discussion

### Comorbidity of perinatal depression, anxiety and PTSD symptoms

Since the individual prevalence rates of depression, anxiety and PTSD symptoms are planned to be published in a separate paper, we only reported the comorbidity rates in this article. Comorbidity of depression and anxiety was more frequently observed in our study (14.5%, 12.1% and 8.1%) at the 6th, 12th and 18 week of postnatal period respectively which is consistent with previous literatures [48,51]. The coexistence of PTSD with depression (9.4%, 6.6% and 3.0% at the 6th, 12th and 18th week of postnatal period respectively) and PTSD with anxiety (9.3%, 6.5% and 3.0% at the 6th, 12th and 18th week of postnatal period respectively) were equally prevalent after childbirth which is in congruent with a previous study [48]. This

**Table 7. Direct, Indirect and total effects of variables associated with comorbidity of depression and PTSD, among postpartum women in Northwest Ethiopia, 2021.**

| Variable's pathway | Type of mental health disorder at each follow up period | | | | | | | | |
|---|---|---|---|---|---|---|---|---|---|
| | T1 Depression & PTSD | | | T2 Depression & PTSD | | | T3 Depression & PTSD | | |
| | Direct effect | Indirect effect | Total effect | Direct effect | Indirect effect | Total effect | Direct effect | Indirect effect | Total effect |
| | β (SE) | β (SE) | β (SE) | β (SE) | β (SE) | β (SE) | β (SE) | β (SE) | β (SE) |
| Direct morbidity Yes (252) | 0.09(0.03) [b] | No path | 0.089(0.028) [b] | 0.04(0.03) [c] | No path | 0.04(0.03)[c] | 0.001(0.02)[c] | No path | 0.001(0.02)[c] |
| Fear of childbirth Yes (245) | 0.003 (0.0004) [a] | 0.001 (0.0001) [a] | 0.003 (0.0004) [a] | 0.001 (0.0003)[b] | 0.001 (0.0001) [a] | 0.002 (0.0004) [a] | -0.0004 (0.0002)[c] | 0.0002 (0.0001) [a] | -0.0002 (0.0002)[c] |
| Gravidity | 0.06(0.02) [b] | 0.01(0.003) [b] | 0.06(0.02) [a] | 0.03(0.02) [c] | 0.01(0.002)[b] | 0.04(0.02)[b] | 0.04(0.01)[b] | 0.002(0.001) [b] | 0.04(0.01) [a] |
| Parity | -0.06(0.02) [b] | No path | -0.06(0.02) [b] | -0.04(0.02)[b] | No path | -0.04(0.02)[b] | -0.04(0.01)[a] | No path | -0.04(0.01)[a] |
| Social support | -0.01(0.006) [b] | -0.02(0.003)[a] | -0.03(0.006) [a] | -0.01(0.005)[c] | -0.01(0.002)[a] | -0.02(0.005)[a] | -0.01(0.004)[b] | -0.002(0.001)[c] | -0.01(0.003)[b] |
| Traumatic birth Yes (306) | 0.19(0.02) [a] | No path | 0.19(0.02) [a] | 0.14(0.02)[a] | No path | 0.14(0.02)[a] | 0.07(0.01)[a] | No path | 0.07(0.01)[a] |
| Indirect morbidity Yes (210) | 0.09(0.03)[b] | 0.07(0.02)[b] | 0.16(0.02)[a] | 0.09(0.027)[a] | 0.03(0.02)[c] | 0.11(0.022)[a] | 0.05(0.02)[b] | 0.001(0.013)[c] | 0.05(0.02)[a] |

[a] p-value ≤0.001

[b] p-value <0.01

[c] p-value >0.05, β is standard estimate.

suggests that women with depression or anxiety are more prone to experience PTSD symptoms which was confirmed by the cross-lagged autoregressive result of this study. The rates of multimorbidity after childbirth were 9.3%, 6.5% and 3.0% at the 6th, 12th and 18th week of postpartum which is consistent with the finding of previous literatures [48,83]. The existence of comorbidities suggests that when a mother is suspected in any of depressive, anxiety or PTSD symptoms, screening for the other disorder symptoms is important for treatment purposes. The results of this study also showed a gradual reduction of comorbidities between depression, anxiety and PTSD symptoms over the 18 weeks of postpartum period. The gradual reduction in comorbidities of depression, anxiety and PTSD indicates the normal adjustments of women over time as reported by previous studies [56,57].

## Cross-lagged autoregressive association of depression, anxiety and PTSD with their comorbidities

Measuring the relation patterns of depression, anxiety and PTSD with their comorbidities over time revealed that all types of comorbidities and multimorbidity predicted the risk of depression and anxiety throughout the follow up period. Ina addition, the PTSD symptom at the second follow up period also predicted the comorbidity of depression and anxiety symptoms at the third follow up period. This confirms the findings of previous researches which reported that depression and anxiety are large components of PTSD and are highly comorbid with PTSD after birth [26,75]. The finding that comorbid anxiety, depression and PTSD symptoms contribute to the persistence of birth related depression and anxiety

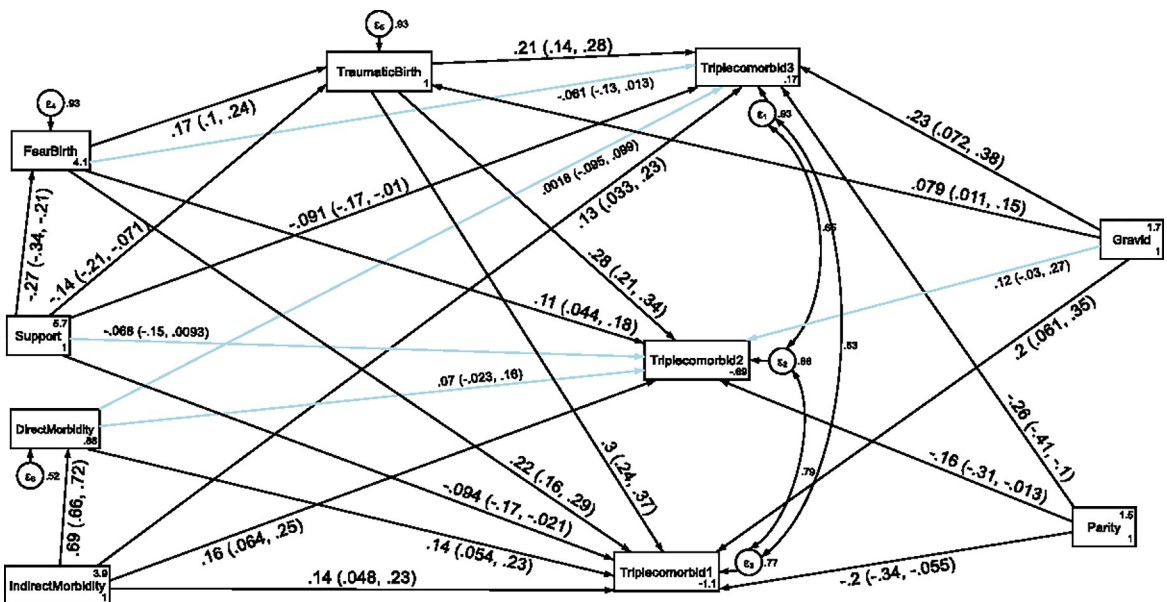

**Fig 9. A structural equation model of the factors associated with triple comorbidity symptoms in postpartum women, Northwest Ethiopia.** Note: β's are standardized estimates with 95% CI.

symptoms in this study is consistent with the finding of another study, which reported that chronicity of individual mental health disorders might be maintained by psychological comorbidity [19].

**Table 8. Direct, indirect and total effects of variables associated with multimorbidity, among postpartum women in Northwest Ethiopia, 2021.**

| Variable's pathway | Type of mental health disorder at each follow up period | | | | | | | | |
|---|---|---|---|---|---|---|---|---|---|
| | T1 Multimorbidity | | | T2 Multimorbidity | | | T3 Multimorbidity | | |
| | Direct effect | Indirect effect | Total effect | Direct effect | Indirect effect | Total effect | Direct effect | Indirect effect | Total effect |
| | β (SE) | β (SE) | β (SE) | β (SE) | β (SE) | β (SE) | β (SE) | β (SE) | β (SE) |
| Direct morbidity Yes (252) | 0.09(0.03) [b] | No path | 0.089(0.028) [b] | 0.04(0.03) [c] | No path | 0.04(0.03) [c] | 0.001(0.02) [c] | No path | 0.001(0.02) [c] |
| Fear of childbirth Yes (245) | 0.003 (0.0004) [a] | 0.001 (0.0001) [a] | 0.003 (0.0004) [a] | 0.001(0.0003) [b] | 0.001 (0.0001) [a] | 0.002 (0.0003) [a] | 0.0004 (0.0002) [c] | 0.0002 (0.0001) [a] | 0.0002 (0.0002) [c] |
| Gravidity | 0.05(0.02) [b] | 0.01(0.003) [b] | 0.06(0.02) [b] | 0.03(0.02) [c] | 0.01(0.002) [b] | 0.03(0.02) [c] | 0.04(0.01) [b] | 0.002(0.001) [b] | 0.04(0.01) [b] |
| Parity | -0.05(0.02) [b] | No path | -0.05(0.02) [b] | -0.04(0.017) [b] | No path | -0.04(0.02) [b] | -0.04(0.01) [a] | No path | -0.04(0.01) [a] |
| Social support | -0.01(0.006) [b] | -0.02(0.003) [a] | -0.03(0.006) [a] | -0.01(0.005) [c] | -0.01(0.002) [a] | -0.02(0.005) [a] | 0.01(0.004) [b] | 0.002(0.001) [c] | 0.01(0.003) [b] |
| Traumatic birth Yes (306) | 0.18(0.02) [a] | No path | 0.18(0.02) [a] | 0.14(0.017) [a] | No path | 0.14(0.02) [a] | 0.07(0.01) [a] | No path | 0.07(0.01) [a] |
| Indirect morbidity Yes (210) | 0.09(0.03) [b] | 0.07(0.02) [b] | 0.16(0.02) [a] | 0.09(0.03) [a] | 0.03(0.018) [c] | 0.14(0.02) [a] | 0.05(0.02) [b] | 0.001(0.013) [c] | 0.05(0.02) [a] |

[a] p-value ≤0.001

[b] p-value <0.01

[c] p-value >0.05, β is standard estimate.

## Direct, indirect and total effects of variables associated with comorbidity of depression, anxiety and PTSD

In this study, direct maternal morbidity, fear of childbirth, gravidity, parity, family size, social support, perceived traumatic childbirth and indirect maternal morbidity were found to have a direct and indirect association with comorbidities of depression, anxiety and PTSD.

Throughout the follow up period, direct maternal morbidity has a direct positive association with comorbidity of depression and anxiety. In addition, it increased the risk of comorbidity between anxiety and PTSD, depression and PTSD and triple comorbidity with a direct positive association at the first follow up period. Indirect maternal morbidity was also found to have a positive indirect association with these events through direct maternal morbidity. This is consistent with previous literatures which have shown that postpartum risks of onset of depression, post-traumatic stress and anxiety are higher among women whose pregnancy included obstetrical complications, compared with women with uneventful pregnancies [49,65,84–86].

Consistent with previous literatures [26,38,47,65], fear of childbirth, gravidity and perceived traumatic childbirth were found to increase the risk of comorbidity between depression, anxiety and PTSD directly and indirectly. In addition, gravidity increased the risk of comorbidity between depression, anxiety and PTSD indirectly through perceived traumatic birth at the three follow up periods. In contrast, multiparity decrease the risk of comorbidity between depression, anxiety and PTSD directly throughout the follow up period. This might be due to the fact that women with large family size may have received more family support than those women with small family size as evidenced by a previous study [56]. Mixed findings were reported for parity including primiparity being a risk factor in one study [87] and multiparity in another study [8]. Therefore, further research is needed to substantiate this. Higher social support also decreases the risk of comorbidity between depression, anxiety and PTSD throughout the follow up period which is in line with previous literatures [1,2,38,47]. It has also an indirect negative effect on the risk of comorbidity between depression, anxiety and PTSD through fear of childbirth and perceived traumatic birth at the three follow up periods. This might be due to the buffering effect of higher social support on negative cognitions since it provides needed social, emotional and physical provisions [1].

## Strength and limitation of the study

Strength of this study is the investigation of longitudinal associations of depression, anxiety and PTSD with their comorbidities using a cross-lagged autoregressive structural equation modeling.

Through a structural equation modelling frame of analysis, the relationships between risk factors for comorbidity of postnatal anxiety, depression and PTSD were examined with adequate sample size and a low attrition rate. This approach allowed us to simultaneously examine the direct and indirect associations between various risk factors with the comorbidity of postnatal anxiety, depression and PTSD. However, this study was not without limitations. Since the study was written and ethically approved before the COVID-19 pandemic, we didn't include the effect this pandemic on the outcome variables of this study. Antenatal factors like depression and anxiety during pregnancy and prior PTSD which may influence the comorbidity rates of these events in the postpartum period were not included in the study. Self-report questionnaires rather than clinical interviews were used to assess anxiety, depression and PTSD which might have inflated the comorbid prevalence rates.

## Conclusion and recommendation

The findings of this study showed that comorbidity of anxiety with depression was the most common among the events of comorbidity between anxiety, depression and PTSD at the 6th, 12th and 18th week of postpartum period. In addition, the coexistence of PTSD with depression, PTSD with anxiety and triple comorbidity was equally prevalent after childbirth throughout the follow up period. Furthermore, investigation of the chronological relations of depression, anxiety and PTSD on their comorbidities indicated that comorbidity of PTSD with depression, PTSD with anxiety and triple comorbidity predicted depression and anxiety in subsequent waves of measurement. While comorbidity of depression and anxiety predicted depression only from T1 to T2, it predicts anxiety in subsequent waves of measurement.

In the current study, direct maternal morbidity, fear of childbirth, higher gravidity, perceived traumatic childbirth and indirect maternal morbidity were also found to have a direct and indirect positive association with comorbidities of depression, anxiety and PTSD. In addition, higher parity, higher family size and higher social support have a direct and indirect negative association with these mental health disorders. Therefore, there should be effective postnatal screening to identify women with psychological problems. Early detection and treatment of anxiety and depression may reduce the likelihood of postnatal psychological disorders. Women with anxiety and depressive symptoms should also be screened for postpartum PTSD symptoms.

Women with direct and indirect maternal morbidities should be identified and treated early, in order to reduce the subsequent burden of comorbid anxiety, depression and PTSD. Adequate information about birth procedures and response to their needs should be given for women with fear of childbirth and perceived traumatic birth to decrease negative emotions of women and fear of childbirth. This would prevent the subsequent comorbid anxiety, depressive and PTSD symptoms. Interventions targeting to encourage social support may also help in increasing mothers' coping and buffer negative cognitions so as to prevent symptoms of comorbid anxiety, depression and PTSD.

## Supporting information

**S1 Dataset. Depression, anxiety and PTSD cross-lagged association dataset.**
(DTA)

## Acknowledgments

The authors would like to acknowledge the heads of Debre Tabor Hospital, Addis Zemen Hospital, Estie Hospital and Nefas Mewcha Hospital for their cooperation on the data collection of this study. The authors are also grateful to the study participants for their dedicated time and volunteer participation.

## Author Contributions

**Conceptualization:** Marelign Tilahun Malaju.

**Data curation:** Marelign Tilahun Malaju.

**Formal analysis:** Marelign Tilahun Malaju.

**Investigation:** Marelign Tilahun Malaju.

**Methodology:** Marelign Tilahun Malaju, Getu Degu Alene, Telake Azale Bisetegn.

**Project administration:** Marelign Tilahun Malaju.

**Resources:** Marelign Tilahun Malaju, Getu Degu Alene, Telake Azale Bisetegn.

**Supervision:** Marelign Tilahun Malaju, Getu Degu Alene, Telake Azale Bisetegn.

**Validation:** Marelign Tilahun Malaju.

**Visualization:** Marelign Tilahun Malaju.

**Writing – original draft:** Marelign Tilahun Malaju.

**Writing – review & editing:** Marelign Tilahun Malaju, Getu Degu Alene, Telake Azale Bisetegn.

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
