## [Decision Letter · Decision Letter 0]

23 May 2022

PONE-D-22-04381Longitudinal path analysis of the relation between comorbidity of anxiety, depression and posttraumatic stress disorder among postpartum women with and without maternal morbidities in Northwest Ethiopia: A cross-lagged autoregressive analysisPLOS ONE

Dear Dr.Marelign Tilahun Malaju ,

Thank you for submitting your manuscript to PLOS ONE. After careful consideration, we feel that it has merit but does not fully meet PLOS ONE’s publication criteria as it currently stands. Therefore, we invite you to submit a revised version of the manuscript that addresses the points raised during the review process.

 The reviewers think that the manuscript needs improvement, particularly in the methods and results. Please address all the comments raised by the reviewers. 

We look forward to receiving your revised manuscript.

Kind regards,

Pracheth Raghuveer, MD, DNB

Academic Editor

PLOS ONE

Journal Requirements:

“This study was conducted as part of PhD degree fulfillment supported by Bahir Dar University and Debre Tabor University. 

The author(s) received no specific funding for this work.”

We note that you have provided additional information within the Funding s Section that is not currently declared in your Funding Statement. Please note that funding information should not appear in the Acknowledgments section or other areas of your manuscript. We will only publish funding information present in the Funding Statement section of the online submission form.

5. Your ethics statement should only appear in the Methods section of your manuscript. If your ethics statement is written in any section besides the Methods, please move it to the Methods section and delete it from any other section. Please ensure that your ethics statement is included in your manuscript, as the ethics statement entered into the online submission form will not be published alongside your manuscript

Reviewers' comments:

Reviewer's Responses to Questions

**Comments to the Author**

1. Is the manuscript technically sound, and do the data support the conclusions?

Reviewer #1: Partly

Reviewer #2: Partly

2. Has the statistical analysis been performed appropriately and rigorously? 

Reviewer #1: Yes

Reviewer #2: I Don't Know

3. Have the authors made all data underlying the findings in their manuscript fully available?

Reviewer #1: Yes

Reviewer #2: Yes

4. Is the manuscript presented in an intelligible fashion and written in standard English?

Reviewer #1: Yes

Reviewer #2: Yes

5. Review Comments to the Author

Reviewer #1: The too many parameters and statistics could not generate new information. In abstract, the percent figures given about coexistence of PTSD with anxiety and triple comorbidity are exactly the same; need rechecking. The statement that "higher social support has a direct or indirect negative association" does not appear true. Limitations of the study, as mentioned by authors, need to be improved to the article useful and solid for future reference.

Reviewer #2: This is a detailed paper on comorbidity of three self-reported psychiatric disorders after complicated delivery, followed up three times after delivery. Thus it is an important paper, but the dense analyses are a bit detailed and difficult to follow for the average clinician who would be expected to translate them into practical suggestions for the patient. My more detailed observations are included in the attached pdf of the paper.

6. PLOS authors have the option to publish the peer review history of their article (what does this mean?). If published, this will include your full peer review and any attached files.

Reviewer #1: **Yes: **Dr. Ramesh Kumar Chandolia

Reviewer #2: No

---

## [Author Response · Author response to Decision Letter 0]

14 Jun 2022

Author’s response for reviewers’ comments

Response for Editor(s)' Comments:

Comments:

Journal Requirements:

Response:

The manuscript is formatted in line with the journal requirement. 

Now the manuscript meets PLOS ONE's style requirements, including those for file naming (see the clean copy and marked copy of the revised document)

Comments:

“This study was conducted as part of PhD degree fulfillment supported by Bahir Dar University and Debre Tabor University. 

The author(s) received no specific funding for this work.”

We note that you have provided additional information within the Funding s Section that is not currently declared in your Funding Statement. Please note that funding information should not appear in the Acknowledgments section or other areas of your manuscript. We will only publish funding information present in the Funding Statement section of the online submission form.

Response: Now, we have removed any funding-related text from the manuscript. 

Comments:

Response: 

Now, we have uploaded the study’s minimal underlying data set as Supporting Information files.

Comments:

4.We note that you have stated that you will provide repository information for your data at acceptance. Should your manuscript be accepted for publication, we will hold it until you provide the relevant accession numbers or DOIs necessary to access your data. If you wish to make changes to your Data Availability statement, please describe these changes in your cover letter and we will update your Data Availability statement to reflect the information you provide.

 Response:

Now, we have already uploaded the minimal dataset as supporting information. Previously I understand the word “repository information for my data at acceptance” as to mean that uploading the supporting information of data file up on acceptance. So, I didn’t understand it clearly. So, please update my Data Availability statement in line with providing minimal dataset as supporting information. 

Comments:

Response:

Now, we have included ethics statement only in the Methods section of the manuscript and removed it from any other section. 

Response for reviewers’ comments 

Comments:

Reviewers' comments:

Reviewer's Responses to Questions

Comments to the Author

1. Is the manuscript technically sound, and do the data support the conclusions?

Reviewer #1: Partly

Reviewer #2: Partly

Response: Now, we have improved the manuscript to make it technically sound, and to make the data supportive of the conclusions. We have revised the manuscript with rigorous literatures and revision. 

2. Has the statistical analysis been performed appropriately and rigorously?

Reviewer #1: Yes

Reviewer #2: I Don't Know

Response: It is shown in the methods and results section. 

3. Have the authors made all data underlying the findings in their manuscript fully available?

Reviewer #1: Yes

Reviewer #2: Yes

Response: It is ok. 

4. Is the manuscript presented in an intelligible fashion and written in standard English?

Reviewer #1: Yes

Reviewer #2: Yes

 Response: It is ok. 

5. Review Comments to the Author

Response for 1st reviewer comments 

Comments:

Reviewer #1: Reviewer #1: The too many parameters and statistics could not generate new information. In abstract, the percent figures given about coexistence of PTSD with anxiety and triple comorbidity are exactly the same; need rechecking. The statement that "higher social support has a direct or indirect negative association" does not appear true. Limitations of the study, as mentioned by authors, need to be improved to the article useful and solid for future reference.

 Response

Now, we have carried out the analysis for the cross-lagged autoregressive and linear structural equation modeling separately to make it clear and understandable for the purpose of doing cross-lagged autoregressive and linear structural equation modeling. The cross-lagged autoregressive modelling is to determine the longitudinal direction of association between depression, anxiety and PTSD. The purpose of linear structural equation modeling is to determine the direct and indirect predictors of comorbidity between depression, anxiety and PTSD. So, that was the purpose of the too many parameters and statistics. See the revised clean and marked copy of the manuscript. 

We have checked the percent figures given about coexistence of PTSD with anxiety and triple comorbidity and they are found to be exactly the same (equal prevalence). 

We have corrected that "higher social support has a direct or indirect negative association” as “higher social support has a direct and indirect negative association”.

Now, we have improved the limitations of the study. See the revised clean and marked copy of the manuscript. 

Response for 2nd reviewer comments 

Comments:

Reviewer #2: This is a detailed paper on comorbidity of three self-reported psychiatric disorders after complicated delivery, followed up three times after delivery. Thus, it is an important paper, but the dense analyses are a bit detailed and difficult to follow for the average clinician who would be expected to translate them into practical suggestions for the patient. My more detailed observations are included in the attached pdf of the paper.

Response:

Now, we have done a separate analysis for the cross-lagged autoregressive and linear structural equation modeling separately to make it clear and understandable for the dense analyses. Doing the analysis separately for the cross-lagged autoregressive and linear structural equation modeling separately could make it simple and clear for the average clinician who would be expected to translate them into practical suggestions for the patient. The cross-lagged autoregressive modelling is to determine the longitudinal direction of association between depression, anxiety and PTSD. The purpose of linear structural equation modeling is to determine the direct and indirect predictors of comorbidity between depression, anxiety and PTSD. So, that was the purpose of the dense analyses. See the revised clean and marked copy of the manuscript. 

Comments

Abstract section: an exceptionally traumatic period due to pandemic with multiple other stresses, so this needs to be mentioned somewhere.

Response

Now, we have included this in the introduction section of the manuscript as follows:

“An exceptionally traumatic period due to COVID-19 pandemic marked by separation, loss of freedom(45), concern about the impact of Covid-19 on pregnancy or possible vertical transmission(46), could be a risk factor for the development of PTSD in postpartum women. There was a recommendation to separate the infant and the mother right after delivery in case of the mother’s positive COVID-19 test(46). These restrictions may lead to a lack of support during and after childbirth, as well as maternal perinatal mental health issues(46)”.

Comments

exposed to what? If it is the factors mentioned below, they could be briefly mentioned here? If it is to COVID19, then the name should be mentioned in the title of the paper itself

Response

Now, we made it clear this in the methods section and removed from the abstract section in order not to exceed the word count of the abstract section when we try to made clarification. It is clarified in the methods section as follows:

Direct and indirect maternal morbidities 

The direct and indirect maternal morbidities were identified based on the WHO maternal morbidity working group criteria(59). According to the WHO maternal morbidity working group criteria, the direct maternal morbidities included in this study were: gestational hypertension, pre-eclampsia, eclampsia, placenta previa, placental abruption, postpartum hemorrhage, mastitis, puerperal sepsis, urinary tract infection, perineal tear, episiotomy wound infection, vaginal wall/perineal laceration and caesarean section wound infection. The indirect maternal morbidities included in this study based on the WHO maternal morbidity working group criteria were: asthma, tuberculosis, influenza, pneumonia, malaria, HIV/AIDS, candidiasis, hepatitis, hypertension, anemia and diabetes mellitus. Women who were diagnosed with any direct and/or indirect maternal morbidities were treated accordingly within the hospitals where they were diagnosed. 

Sampling procedure 

Women diagnosed with any of the direct and indirect maternal morbidities were taken as exposed

mothers and included in the study. Women without the direct and indirect maternal morbidities were taken as non-exposed mothers. All women with direct maternal morbidities were included in the study and women without direct maternal morbidities were selected by simple random sampling method using their chart number on daily bases. The chart numbers of women without direct maternal morbidities were entered into computer to generate random numbers using Microsoft Excel for random selection of women. The recruitment of women continued prospectively until the required sample size was fulfilled. Women were asked to give written consent to participate in the study and after getting their consent and full address, appointments were made at their home to collect the data for the follow up study. 

Comments

were any clinical criteria or a diagnostic instrument used apart from the WHO morbidity criteria?

Response

We only used the WHO morbidity criteria and it is clarified in the methods section as follows:

Direct and indirect maternal morbidities 

The direct and indirect maternal morbidities were identified based on the WHO maternal morbidity working group criteria(59). According to the WHO maternal morbidity working group criteria, the direct maternal morbidities included in this study were: gestational hypertension, pre-eclampsia, eclampsia, placenta previa, placental abruption, postpartum hemorrhage, mastitis, puerperal sepsis, urinary tract infection, perineal tear, episiotomy wound infection, vaginal wall/perineal laceration and caesarean section wound infection. The indirect maternal morbidities included in this study based on the WHO maternal morbidity working group criteria were: asthma, tuberculosis, influenza, pneumonia, malaria, HIV/AIDS, candidiasis, hepatitis, hypertension, anemia and diabetes mellitus. Women who were diagnosed with any direct and/or indirect maternal morbidities were treated accordingly within the hospitals where they were diagnosed. 

Comments

insert postnatal here 

Response

It is accepted and written as follows in the abstract section:

Results: Comorbidity of anxiety with depression was also the most common (14.5%, 12.1% and 8.1%) at the 6th, 12th and 18th week of postnatal period respectively. 

Comments

authors might mention briefly the presence or absence of clinical intervention as an intervening variable. 

Response

Now, we have briefly mentioned the presence or absence of clinical intervention in the methods section but we didn’t consider it as an intervening variable because treating all mothers with any morbidity is a must and if all are treated then, since there will be no difference between absence and presence of intervention, no need to consider it as a variable. 

We briefly indicated this issue as follows:

Direct and indirect maternal morbidities 

The direct and indirect maternal morbidities were identified based on the WHO maternal morbidity working group criteria(59). According to the WHO maternal morbidity working group criteria, the direct maternal morbidities included in this study were: gestational hypertension, pre-eclampsia, eclampsia, placenta previa, placental abruption, postpartum hemorrhage, mastitis, puerperal sepsis, urinary tract infection, perineal tear, episiotomy wound infection, vaginal wall/perineal laceration and caesarean section wound infection. The indirect maternal morbidities included in this study based on the WHO maternal morbidity working group criteria were: asthma, tuberculosis, influenza, pneumonia, malaria, HIV/AIDS, candidiasis, hepatitis, hypertension, anemia and diabetes mellitus. Women who were diagnosed with any direct and/or indirect maternal morbidities were treated accordingly within the hospitals where they were diagnosed. 

Comments

Also, PTSD due to what? 

Response

 Now, we have indicated the PTSD due to what? In the abstract section as follows:

“…comorbidity of PTSD (due to perceived traumatic birth) with depression, PTSD with anxiety…”

Introduction

Comments: 

perhaps this working definition could be mentioned in the abstract as well?

Response:

Now, we have mentioned it in the abstract section as follows:

“…Traumatic birth has been defined as an event occurring during labour and birth that may be a serious threat to the life and safety of the mother and/or child…”

Comments: 

could it also be due to the type of clinical care or lack thereof, during and after pregnancy?

Response:

Now, we have included this as follows:

“…During the postpartum period, women can develop PTSD symptoms due to a difficult or traumatic births such as; emergency cesarean section or instrumental deliveries during which women think that they or their baby might die or be seriously hurt (31-37) …”

Comments:

re write- 'to the authors knowledge, there is no Ethiopian study....)

Response:

Now, we have accepted and re-written it as follows:

“…To the best of the authors’ knowledge, except for the prevalence of postpartum depression, there is no study which have investigated the prevalence of postpartum anxiety, PTSD, comorbidity and multimorbidity in Ethiopia…”

Methods section:

Comments:

how was the sample size determined?

Response:

Now, we have included how the sample size was determined and indicated in the methods section as follows:

Sample Size determination

Sample size was calculated using a two-population proportion formula with Epi-Info version 7. Accordingly, a minimum sample size of 746 (249 exposed and 497 non-exposed) was calculated by taking 0.05 alpha (α), power of 90 %, odds ratio of 4.23, 2.3% of mothers without depression during pregnancy who develop PTSD in postpartum period, 1:3 ratio of exposed to non-exposed (since the controls were 3 times the cases in a previous study(58)) and by adding 10% non-response rate. These parameters used for the sample size calculation were taken from a previous study(58).

Comments

out of a total of how many women registered for pregnancy at these hospitals? Were those who consented, similar or different from those who did not in the total population of women registered and delivering at these hospitals? 

Response:

Now, we have clarified this issue in the methods section as follows:

Study population 

A total of 775 women consented to participate in the study and participated at the first, second and third follow-up of the study (6th, 12th and 18th week of postpartum period). Recruitment of the study participants was done after child birth and before the time of discharge within the hospitals where women gave birth. All women who were asked for consent agreed to participate in the study. The selected women were among those who were attending their antenatal care (ANC) in the study hospitals and among those who came for delivery in the study hospitals from different health centers and/or hospitals through referral. Therefore, the total number of delivering women in the study hospitals were not known before the initiation of the study. 

Comments:

was any limit placed on the duration of the pre-term? Eg first or second trimester miscarriages?

Response:

Now, we have clarified this issue in the methods section as follows:

Eligibility/Inclusion Criteria 

Women aged 15 years and above, with preterm birth (between 28 -36 weeks), term or post term delivery and with live birth, still birth or fetal death were included in the study. Women who are unable to communicate (having hearing problem and cannot communicate with sign language) were excluded from the study. 

 Comments:

in the introduction, it might be educative to mention the Ethiopian population rates of the common diseases mentioned here such as anemia, malaria, hypertension, asthma, tuberculosis, HIV, diabetes mellitus. 

Response:

Now, we have included this issue in the introduction section as follows:

“…While the prevalence of gestational diabetes mellitus was reported to be 12.8% in Ethiopia(53), magnitude of other medical illnesses among child delivering Ethiopian women were also reported to be 40% (anemia), 5% (HIV), 3% (Tuberculosis) and 2% (Malaria)(54)…”

Comments:

Authors should provide number of items, scoring pattern, cutoffs and time taken for each scale they used. What are the minimum and maximum scores for each sub-scale, and the cutoff as given by the authors of the scale? Also were the scores calculated on subscales or the entire scale score? Does the scale actually measure 'stress' or was this done for the present study only? What should the duration of symptoms be for possible ''caseness'', as per this scale?

Response:

Now, we have provided it in the methods section as follows:

Depression, anxiety and stress

The short version of depression, anxiety and stress scale 21 (DASS-21) questionnaire was used to measure depression, anxiety and stress. DASS-21 is a psychological screening instrument which is capable of differentiating symptoms of depression, anxiety and stress. It is a validated and reliable instrument with 21 items in three domains. Each domain comprises seven items assessing symptoms of depression, anxiety and stress. Participants were asked to indicate the presence of symptoms in each domain over the past week scoring from 0 (did not apply at all). to 3 (applied most of the time). Scores from each dimension were summed. Then, the final score was multiplied by 2 and then categorized according to the DASS manual as normal, mild, moderate, severe and extremely severe. Accordingly, for participants with depression, a depression score of 0–9 is considered normal, 10–13 as mild, 14–20 as moderate, 21–27 as severe and 28 and above as extremely severe. In this study a score ≥ 10 was considered for a mother to have a symptom of depression. For participants with anxiety, an anxiety score of 0–7 is considered normal, 8–9 as mild, 10–14 as moderate, 15–19 as severe and 20 and above as extremely severe. A cut-off score of ≥ 8 was considered to have symptoms of anxiety for this study. For participants with stress, a stress score of 0–14 is considered normal, 15–18 as mild, 19–25 as moderate, 26–33 as severe and 34 and above as extremely severe. A score of ≥ 15 was considered to have symptoms of stress for this study. This instrument was validated and used previously in Ethiopia (60, 61). 

Comments:

At which point was informed consent obtained? Did these personnel receive any training in scale administration to avoid bias? Did the same group administer the scales or different people? What were their qualifications? Where were the scales administered to ensure confidentiality? Where did the follow-up interviews take place? Same questions as above?

Response:

Now, we have clarified this issue in the methods section as follows:

“…Two different groups of data collectors were used for data collection to avoid bias. The first group were health professionals with BSC degree in Nursing and Midwifery who collected the baseline data and the direct and indirect maternal morbidities based on the WHO maternal morbidity working group criteria. The second group were health extension workers (community health workers) who collected the follow up data by house-to-house visit (home visit). Supervision was done by the principal investigator. Training was given for data collectors in scale administration. During the training process, data collectors carefully reviewed each question and conduct pretest before the study commenced. The investigator and data collectors checked the questionnaire during the pretest and amendment was done as required.”

Ethical Consideration

“…. Each study participant has given written informed consent to participate in the study before taking part after giving birth and before discharge…”

Comments

Commenced

Response:

This is accepted and corrected

Comments:

what about all the other scales? Were they incorporated in this model or not? If not then do the authors plan to publish them in a separate paper (which is acceptable, but should be mentioned here).

Response:

This is clarified as follows:

“…The Autoregressive Cross-lagged (ARCL) modeling strategy was used to examine the longitudinal association of depression, anxiety and PTSD with their comorbidities including all other scales used for this study…”

Comments:

how was autoregression calculated? again, how does prediction work?

Response:

This is computer intensive using STATA software by using cross-lagged autoregressive analysis and linear structural equation modelling. Now, we have tried to clarify is as follows:

Data processing and analysis

We used a three-wave, cross-lagged autoregressive structural equation modeling. The analysis was carried out by using Stata version 16 software. The Autoregressive Cross-lagged (ARCL) modeling strategy was used to examine the longitudinal association of depression, anxiety and PTSD with their comorbidities including all other scales used for this study. This modeling strategy incorporates three main components. First, the stability/autoregressive effects (e.g., effects of T1 comorbid depressive, anxiety and PTSD symptoms on their respective T2 variables). That means, later measures of a construct are predicted by earlier measures of the same construct. Second, the cross-lagged effects (e.g., effect of T1 depressive symptoms on T2 comorbid PTSD and anxiety symptoms and of T1 comorbid PTSD and anxiety symptoms on T2 depressive symptoms). That means, earlier measures of depression predict later measures of comorbid PTSD and anxiety symptoms. This model can be extended to examine bi-directional relations such that earlier measures of PTSD predict later measures of comorbid depression and anxiety as well. Third, the synchronous associations between the unexplained variances of these variables at T1, T2 and T3(74, 75). 

Comments:

Since the study was done at three centers, all 3 ethics committees should have been approached for permission. Was that done?

Response:

Now, we have clarified this as follows:

“…All institutions of the study area were also approached for permission…”

Comments:

Was this witnessed by a neutral observer to prevent coercion?

Response:

Now, we have clarified this as follows:

“…Data collectors read out and assisted participants to fill out the consent form if participants were unable to read and write. This was witnessed by a neutral observer to prevent coercion…”

Results section

Comments:

Could a justification for including those without risk (ie controls) to be double the number of those who suffered risk (cases) be provided with the sampling procedure? Also were the cases and controls comparable? Were there any significant differences between the two samples as also the total number registered and delivering at these hospitals?

Response:

Now, we clarified this in the methods section as follows: 

Sample Size determination

“…1:3 ratio of exposed to non-exposed (since the controls were 3 times the cases in a previous study(58))…”

Study population 

“…Women differ in their socio-demographic characteristics, reproductive, obstetrics and medical variables. The effect of these variations on the outcome variables were controlled with the use of multivariable analysis in this study (multivariable linear structural equation modelling) …”

Comments:

What about single morbidity only? Could a column be included here? Or did the authors ONLY include those with comorbidities of anxiety and depression (traumatic experience and PTSD thereof being an independent factor), otherwise how did they separate out independent anxiety and depression symptoms vs. their being part of the PTSD itself?

Response:

Now, we have included this as follows:

Individual prevalence rates of depression, anxiety and PTSD symptoms

The prevalence of depression, anxiety and PTSD symptoms at the 6th, 12th and 18th week of postpartum period was computed and is provided in Table 2. The most common disorder was anxiety followed by depression throughout the follow up period. PTSD symptom was rarely reported at each time point. Stressor criteria A for a traumatic birth were fulfilled by 37.03% of women using DSM-IV criteria and 39.5% using DSM-5 criteria. Using DSM-5 stressor criteria therefore increased the number of women identified as fulfilling stressor criteria by 2.47%. 

Comments:

this term needs explanation- indirect maternal morbidity 

Response:

Now, we have clarified this as follows:

In addition, indirect maternal morbidity (asthma, tuberculosis, pneumonia, hypertension, anemia and diabetes mellitus) had also an indirect positive effect on comorbidity of anxiety and PTSD symptoms at the first and second follow up period through direct maternal morbidity.

Discussion:

Comments:

results do need to include single factor morbidity too, as difficult childbirth is a known risk factor for mental morbidity.

Response:

Now, we have clarified this as follows:

“…Since the individual prevalence rates of depression, anxiety and PTSD symptoms are planned to be published in a separate paper, we only reported the comorbidity rates in this article…”

Comments:

is larger family size a proxy for greater support to the mother and child?

Response:

Now, we have clarified this as follows:

“…In contrast, multiparity decrease the risk of comorbidity between depression, anxiety and PTSD directly throughout the follow up period. This might be due to the fact that women with large family size may have received more family support than those women with small family size as evidenced by a previous study(56)…”

Comments:

Authors describe a number of other risk factors in their abstract which they do not discuss either in results or discussion sections. This should be done. 

Response:

Now, we have included risk factors written in the abstract to the results and discussion section.

Comments:

Also, some simplification of results to make them more perceptible to the clinician would be useful. 

Response:

Now, we have done a separate analysis for cross-lagged autoregressive analysis and linear structural equation modelling separately for simplification of results. 

Comments:

Authors also did not comment on the gradual REDUCTION of co-morbidity over time.

Response: 

Now, we have clarified this issue as follows:

“…The results of this study also showed a gradual reduction of comorbidities between depression, anxiety and PTSD symptoms over the 18 weeks of postpartum period. The gradual reduction in comorbidities of depression, anxiety and PTSD indicates the normal adjustments of women over time as reported by previous studies (56, 57)…”

I thank you very much!

---

## [Decision Letter · Decision Letter 1]

4 Aug 2022

Longitudinal path analysis for the directional association of depression, anxiety and posttraumatic stress disorder with their comorbidities and predictors among postpartum women in Northwest Ethiopia: A cross-lagged autoregressive modelling study

PONE-D-22-04381R1

Dear Dr. Marelign Tilahun Malaju

We’re pleased to inform you that your manuscript has been judged scientifically suitable for publication and will be formally accepted for publication once it meets all outstanding technical requirements.

Kind regards,

Pracheth Raghuveer, MD, DNB

Academic Editor

PLOS ONE

Additional Editor Comments (optional):

Reviewers' comments:

Reviewer's Responses to Questions

**Comments to the Author**

1. If the authors have adequately addressed your comments raised in a previous round of review and you feel that this manuscript is now acceptable for publication, you may indicate that here to bypass the “Comments to the Author” section, enter your conflict of interest statement in the “Confidential to Editor” section, and submit your "Accept" recommendation.

Reviewer #2: All comments have been addressed

2. Is the manuscript technically sound, and do the data support the conclusions?

Reviewer #2: Yes

3. Has the statistical analysis been performed appropriately and rigorously? 

Reviewer #2: Yes

4. Have the authors made all data underlying the findings in their manuscript fully available?

Reviewer #2: Yes

5. Is the manuscript presented in an intelligible fashion and written in standard English?

Reviewer #2: Yes

6. Review Comments to the Author

Reviewer #2: No comments. If, in addition to their replies to reviewers, the authors had also highlighted their revisions in the main body of ther revised submission, it would be easier for the reviewer to check new insertions and revisions of text.

7. PLOS authors have the option to publish the peer review history of their article (what does this mean?). If published, this will include your full peer review and any attached files.

Reviewer #2: **Yes: **Smita Neelkanth Deshpande

---

## [Editor Report · Acceptance letter]

5 Aug 2022

PONE-D-22-04381R1 

Longitudinal path analysis for the directional association of depression, anxiety and posttraumatic stress disorder with their comorbidities and predictors among postpartum women in Northwest Ethiopia: A cross-lagged autoregressive modelling study 

Dear Dr. Malaju:

I'm pleased to inform you that your manuscript has been deemed suitable for publication in PLOS ONE. Congratulations! Your manuscript is now with our production department. 

Kind regards, 

on behalf of

Dr. Pracheth Raghuveer 

Academic Editor

PLOS ONE